# Type 1 interferons and Foxo1 down-regulation play a key role in age-related T-cell exhaustion in mice

Aurélie Durand[1], Nelly Bonilla[1], Théo Level[1], Zoé Ginestet [1], Amélie Lombès [1], Vincent Guichard[1], Mathieu Germain[1], Sébastien Jacques[1], Franck Letourneur [1], Marcio Do Cruzeiro[1], Carmen Marchiol[1], Gilles Renault [1], Morgane Le Gall [1], Céline Charvet [2,3,4,5], Agnès Le Bon [1], Bruno Martin[1], Cédric Auffray[1] & Bruno Lucas [1] ✉

Foxo family transcription factors are critically involved in multiple processes, such as metabolism, quiescence, cell survival and cell differentiation. Although continuous, high activity of Foxo transcription factors extends the life span of some species, the involvement of Foxo proteins in mammalian aging remains to be determined. Here, we show that Foxo1 is down-regulated with age in mouse T cells. This down-regulation of Foxo1 in T cells may contribute to the disruption of naive T-cell homeostasis with age, leading to an increase in the number of memory T cells. Foxo1 down-regulation is also associated with the up-regulation of co-inhibitory receptors by memory T cells and exhaustion in aged mice. Using adoptive transfer experiments, we show that the age-dependent down-regulation of Foxo1 in T cells is mediated by T-cell-extrinsic cues, including type 1 interferons. Taken together, our data suggest that type 1 interferon-induced Foxo1 down-regulation is likely to contribute significantly to T-cell dysfunction in aged mice.

Aging of an organism is progressive and cumulative, leading to alterations in the functions of multiple cells, tissues, and organs[1]. Aging is a complex process that affects many aspects of mammalian biology, including the immune system. Indeed, age-related alterations in the immune system make older adults more susceptible to infectious diseases and tumors, resulting in increased morbidity and mortality[2–4]. In addition, the effectiveness of vaccination is significantly reduced in the older population, limiting preventive prophylaxis[5,6]. Because of these age-associated defects, immunologists have become interested in understanding the aging process[7–9]. Although aging affects both the innate and adaptive immune systems, these changes are most pronounced in adaptive immune cells[10,11].

Many mutations in the insulin/IGF-1 pathway extend life span in worms, flies, and mammals[12]. This pathway is highly influenced by the transcription factors of the Forkhead box O family (Foxo)[12,13]. Foxo genes are critically involved at the crossroads of different processes, such as metabolism, quiescence, cell survival, oxidative stress resistance, cellular differentiation, autophagy and apoptosis[14,15]. Over the last few decades, studies in various model organisms have highlighted the crucial role of Foxo family members in preventing aging and extending life span. First, in *Caenorhabditis elegans*, increased activity of daf-16 which encodes a Foxo family transcription factor doubles their life span[16,17]. Second, in *Drosophila melanogaster*, Foxo over-expression also extends the fly's life span[18,19]. Finally, in non-senescent *Hydra vulgaris*, continuously high activity of transcription factor Foxo contributes to continuous stem cell proliferation[20,21]. However, the involvement of Foxo proteins in mammal longevity as well as in the aging of the immune system has not yet been documented.

[1]Université Paris-Cité, Institut Cochin, Centre National de la Recherche Scientifique (CNRS) UMR8104, Institut National de la Santé et de la Recherche Médicale (INSERM) U1016, 75014 Paris, France. [2]Institut de Génétique et de Biologie Moléculaire et Cellulaire (IGBMC), Illkirch, France. [3]CNRS UMR7104, Illkirch, France. [4]INSERM U1258, Illkirch, France. [5]Université de Strasbourg, Strasbourg, France. ✉e-mail: bruno.lucas@inserm.fr

Over the last decade, it has been shown that Foxo transcription factors (especially Foxo1 and Foxo3, the two members of this family expressed by T cells) control many aspects of T-cell physiology in mammals. They increase T-cell survival by inducing IL-7 receptor expression and T-cell trafficking by increasing the expression of key molecules involved in their entry and exit from lymphoid and non-lymphoid tissues such as CD62L, S1PR1, and CCR7[22,23]. In addition, Foxo1 plays a crucial role in directing the differentiation of naive CD4 and CD8 T cells into effector cells[24–28].

In this paper, we show that Foxo1 expression is progressively down-regulated in mouse T cells with age and that this decrease imprints their transcriptional signature. Comparison of the T-cell compartment of old wild-type (WT) mice (22 months) with that of young adult mice (3 months) deficient or proficient in Foxo1 expression in T cells allowed us to assess the full consequences of the loss of Foxo1 expression on T-cell aging. Finally, as adoptive transfer of T cells from young adults to old animals leads to their accelerated aging, we then sought to identify the T-cell extrinsic factors leading to the decrease of Foxo1 expression in T cells and to their subsequent aging.

## Results

### Gradual down-regulation of Foxo1 in T cells with age imprints their transcriptional signature

Throughout this study, as in our previous articles[29,30], regulatory CD4 T cells (CD4$_R$) were defined as Foxp3$^+$ CD4$^+$ CD8α$^-$ TCRβ$^+$ cells, memory CD4 T cells (CD4$_M$) as CD44$^{hi}$ Foxp3$^-$ CD4$^+$ CD8α$^-$ TCRβ$^+$ cells and naive CD4 T cells (CD4$_N$) as CD44$^{-/low}$ Foxp3$^-$ CD8α$^-$ TCRβ$^+$ cells (Fig. S1a). CD44 expression was also used to discriminate between naive and memory CD8 T cells (CD8$_N$ and CD8$_M$ respectively, Fig. S1a). Foxo1 and Foxo3 were differentially expressed in the various T-cell subsets in secondary lymphoid organs (SLOs) of young adult mice with, for example, CD8$_N$ and CD8$_M$ T cells expressing more Foxo1 than CD4$_N$, CD4$_M$, and CD4$_R$ T cells (Fig. S1b). To assess changes in the expression of Foxo1 and Foxo3 with age, we next compared peripheral T cells from young adult and old mice (3-month-old versus 22-month-old). For this purpose, cell suspensions from SLOs of young adult and old mice were first labeled separately with anti-CD45 antibodies conjugated to different fluorochromes and then mixed together so that surface and intracellular staining were performed in a single well. Interestingly, Foxo1 expression decreased significantly in all T-cell subsets with age, whereas Foxo3 expression decreased slightly in CD4$_M$ and CD8$_M$ cells only (Figs. 1a, S1c). Foxo1 down-regulation with age was progressive with levels already decreased in blood T cells from 6-month-old mice compared with similar cells from 3-month-old mice (Fig. 1b).

When phosphorylated at threonine 24, serine 256, and serine 319 by AKT, Foxo1 is translocated from the nucleus to the cytoplasm and then addressed to the proteasome and degraded[31]. To investigate whether such a process might be involved in the down-regulation of Foxo1 with age, we next studied the localization of Foxo1 in T cells (Fig. S1d). Foxo1 expression decreased with age to a similar extent in the nucleus and the cytoplasm of naive, memory, and regulatory CD4 T cells from old mice (Fig. S1e). Thus, in T cells from old mice, we did not observe a decrease in Foxo1 in the nucleus to the benefit of the cytoplasm since the contribution of nuclear Foxo1 to total Foxo1 staining was even slightly increased with age (Fig. S1e). However, as Foxo1 down-regulation is very progressive with age, Foxo1 degradation when excluded from the nucleus could prevent its accumulation in the cytoplasm.

We then assessed whether the AKT signalization pathway could be involved in Foxo1 down-regulation with age. Indeed, as indicated above, AKT phosphorylates Foxo1, leading to its translocation from the nucleus to the cytoplasm[31]. The amount of phosphorylated Foxo decreased in all T-cell subsets with age, but these results may only reflect the down-regulation of Foxo1 with age (Fig. 1c. Fluorescence

minus one controls are shown in Fig. S2a). Consistent with the literature[32], we found that the extent of Foxo phosphorylation correlated significantly with the extent of AKT phosphorylation in CD4$_N$, CD4$_R$, CD4$_M$, and CD8$_M$ cells and tended to correlate with the extent of AKT phosphorylation in CD8$_N$ cells (Fig. 1d). AKT activation may thus contribute to Foxo1 phosphorylation in T cells from old mice. However, phosphorylated AKT increased significantly with age only in CD4$_N$ and CD8$_N$ cells (Fig. 1c). Interestingly, in old mice, the amount of phosphorylated AKT in these 2 T-cell subsets correlated significantly with the extent of Foxo1 down-regulation suggesting that increased AKT activation may play a role in Foxo1 down-regulation with age at least in these cells (Fig. S2b).

We then studied the transcriptome of CD4$_N$ cells as a function of mouse age. We decided to focus on CD4$_N$ cells as naive T cells can be considered as more homogenous than CD4$_M$ and CD4$_R$ cells. Of note, half of the genes were up-regulated with age and half down-regulated (Fig. 1e). In agreement with the results obtained at the protein level, we found that transcription of Foxo1, but not that of Foxo3 was diminished in CD4$_N$ cells from 22-month-old mice compared to CD4$_N$ cells from 3-month-old mice (Fig. 1f). To assess whether Foxo1 down-regulation with age in CD4$_N$ cells impacted their transcriptional signature, we next determined the transcriptomic signature of CD4$_N$ cells lacking the expression of Foxo1 (CD4$_N$ cells from CD4$^{cre}$ Foxo1$^{fl/fl}$ mice alias Foxo1$^{TKO}$ mice[27,28] compared to CD4$_N$ cells from Foxo1$^{fl/fl}$ mice not expressing the Cre recombinase alias Foxo1$^{Ctrl}$ mice; in Foxo1$^{TKO}$ mice, Cre recombinase-mediated Foxo1 DNA recombination occurs at the double-positive (CD4$^+$CD8$^+$) stage of thymic differentiation and thus affects both CD4 and CD8 T-cell lineages). We found 202 up-regulated genes (fold change > 1.5; $p < 0.05$) and 144 down-regulated genes (fold change < −1.5; $p < 0.05$) in CD4$_N$ cells from the lymph-nodes of Foxo1$^{TKO}$ mice (Supplementary Data 1) compared to control littermates (Foxo1$^{Ctrl}$ mice). Interestingly, genes down-regulated in CD4$_N$ cells from Foxo1$^{TKO}$ mice were mostly less expressed by CD4$_N$ cells from old WT mice (64.6%; Fig. 1g). Similarly, most genes overexpressed in CD4$_N$ cells from Foxo1$^{TKO}$ mice were found to be up-regulated in CD4$_N$ cells from old WT mice (72.8%; Fig. 1g). This bias was even reinforced when fold changes were defined as 2 and −2 respectively (Fig. S2c). Gene Set Enrichment Analysis (GSEA[33,34];) confirmed a clear and significant overlap between the transcriptional signatures of CD4$_N$ cells from young adult Foxo1$^{TKO}$ and old WT mice (Figs. 1h, S2d). Finally, we validated our transcriptomic results at the protein level in CD4$_N$ and CD8$_N$ cells from old WT mice by showing that the expression of CCR7 and CD62L which is known to be controlled by Foxo1[22,23], was closely correlated with their level of expression of Foxo1 (Fig. S2e–j). Thus, Foxo1 down-regulation with age in naïve T cells is sufficient to significantly reshape their transcriptional signature.

### The peripheral T-cell compartments of young adult Foxo1$^{TKO}$ mice and old wild-type mice share many features

To assess the full consequences of Foxo1 down-regulation with age on T cells, we then decided to thoroughly compare the peripheral T-cell compartment of aged WT mice with that of young adult Foxo1$^{TKO}$ mice (Fig. 2a). In peripheral lymph-nodes, mesenteric lymph-nodes and the spleen of old WT and young adult Foxo1$^{TKO}$ mice, we observed a marked increase in the proportions of CD4$_M$ and CD4$_R$ cells at the expense of CD4$_N$ cells when compared to their respective control mice (Fig. 2b). Reduced proportions of CD8$_N$ cells were also found in the SLOs of old WT mice and young adult Foxo1$^{TKO}$ mice (Fig. S3a). These changes in the proportions of the various peripheral T-cell subsets resulted at least in part from a decrease in the absolute number of naive T cells in all SLOs with age or in the absence of Foxo1 expression (Figs. 2c, S3b). Absolute numbers of CD4$_M$ cells were markedly increased in the SLOs of old WT and young adult Foxo1$^{TKO}$ mice whereas those of CD8$_M$ cells were augmented in all SLOs only in old WT mice (Figs. 2c, S3b). Changes in absolute CD4$_R$ cell counts varied by

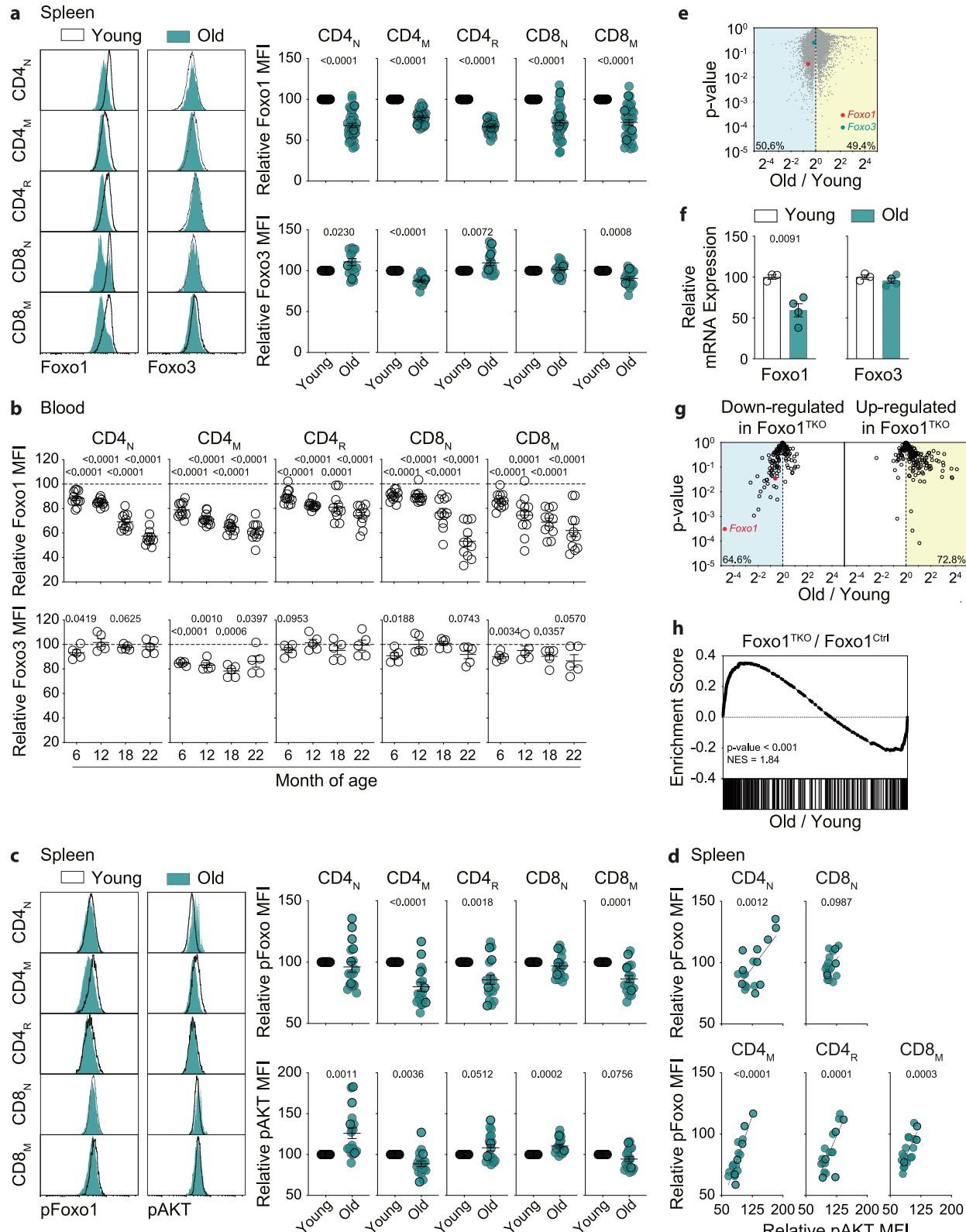

SLOs, with for example an increase in the spleen and a decrease in peripheral lymph-nodes of both old WT and young adult Foxo1$^{TKO}$ mice. Overall, the peripheral CD4 T-cell compartment of old WT mice and young adult Foxo1$^{TKO}$ mice is very similar in terms of proportions and absolute numbers of CD4$_N$, CD4$_M$, and CD4$_R$ cells. By contrast, the similarity of the CD8 T-cell compartment from the SLOs of the two types of mice results mainly from reduced absolute numbers of CD8$_N$ cells.

It has been shown by us and others that Foxo1 restrains CD4$_N$-cell differentiation into type 17 Helper (TH17)[27] and follicular helper (TFH) T cells[26]. To characterize more deeply memory/effector T cells from old WT mice, we next determined the proportions of these two subsets among CD4$_M$ cells as a function of mouse age. TFH cells were defined as PD1$^+$ CXCR5$^+$ CD4$_M$ cells and TH17 as CD4$_M$ cells producing IL-17 upon in vitro restimulation (Fig. S3c, d). We found that at least in the peripheral lymph nodes of both old WT mice and young adult Foxo1$^{TKO}$

**Fig. 1 | Foxo1 is gradually down-regulated in T cells with age. a** Cell suspensions from the spleen of a 3-month-old mouse (Young adult) and a 22-month-old mouse (Old) were first stained separately with anti-CD45 antibodies conjugated to different fluorochromes and then mixed before further staining. Foxo1 and Foxo3 fluorescence histograms of the indicated T-cell subsets are shown for representative mice (left panel). Relative Mean Fluorescence Intensities (MFIs) were calculated by dividing the MFI of a given T-cell subset of the old mouse by the MFI of the same T-cell subset of the barcoded young adult mouse (right panel). **b** A cohort of female mice was bled at different ages and blood cells were mixed with cells from 3-month-old female mice before revealing Foxo1 and Foxo3 expression. Of note, before mixing them, blood cells were barcoded with anti-CD45 antibodies conjugated to different fluorochromes. Relative MFIs are shown for the indicated T-cell subsets. **c** phospho-Foxo (pFoxo) and phospho-AKT (pAKT) fluorescence histograms of the indicated T-cell subsets are shown for representative mice (left panel). Relative MFIs were calculated by dividing the MFI of a given T-cell subset of the old mouse by the MFI of the same T-cell subset of the barcoded young adult mouse (right panel). **d** Relative expression of pFoxo by the indicated T-cell subsets from the spleen of old mice as a function of pAKT relative expression. **e** Volcano plot representation of the transcriptomic signature of $CD4_N$ cells from aged versus young adult mice. **f** The relative expression of Foxo1 and Foxo3 mRNAs was directly derived from the obtained signature and was plotted as Mean ± SEM ($n = 3$ Young adult mice; $n = 4$ old mice). **g, h** A list of genes up- and down-regulated in $CD4_N$ cells from Foxo1[TKO] mice compared with $CD4_N$ cells from Foxo1[Ctrl] mice was also determined ($p$ value of < 0.05 and fold change > 1.5 or <−1.5, see Supplementary Data 1). The expression of these genes by $CD4 T_N$ cells as a function of mouse age was visualized using the Volcano (**g**) or GSEA (**h**) plot. Quantifications are represented as Means ± SEM. The significance of differences between two series of results was assessed using Student's unpaired $t$ test (**a**–**c**, **f**). For assessing correlations, Pearson correlation (two-sided test, coefficient, and 95% confidence intervals) was used (**d**). The p-values for the GSEA test statistics are calculated by permutation (**h**). Significant ($p < 0.05$) or almost significant ($0.05 < p < 0.10$) p-values are indicated. Source data are provided as a Source Data file.

mice, the proportions of TH17 and TFH among $CD4_M$ cells were significantly increased. Thus, the peripheral T-cell compartments of old WT mice and young adult Foxo1[TKO] mice share not only quantitative similarities in terms of proportions of naïve, memory, and regulatory T cells but also qualitative similarities.

The reduction in the absolute numbers of naive T cells in SLOs with age results, at least in part, from the decreased production of naive T cells in the thymus due to its involution[35]. To assess whether the reduction in the numbers of naive T cells in the SLOs of Foxo1[TKO] mice could also result from thymic dysfunction, we next studied the thymi of 4-week-old Foxo1[TKO] mice and showed that they contained as many CD4⁻CD8⁻TCR$_\beta$⁻ triple-negative, CD4⁺CD8⁺ double-positive, $CD4_N$ and $CD8_N$ cells as the thymi of their control littermates. (Fig. 2d). Analysis of the wave of new thymic migrants reaching SLOs 16 h after intra-thymic injection of fluorescein revealed that while the numbers of FITC⁺ $CD4_N$ and $CD8_N$ cells were normal in the spleen, they were significantly lower in the lymph-nodes of Foxo1[TKO] mice compared with Foxo1[Ctrl] mice (Fig. 2e). In agreement with these results, at 4 weeks of age, the absolute numbers of naive T cells were reduced in the lymph-nodes but not in the spleen of Foxo1[TKO] mice (Fig. S4a). As previously described by Kerdiles et al.[23], we observed that naïve T cells from Foxo1[TKO] mice expressed low levels of CD62L and CCR7 compared to their T-cell counterparts from Foxo1[Ctrl] mice (Fig. S4b, c). As these 2 molecules are involved in the entry of T cells into lymph nodes, their down-regulation may account for the decreased naïve T-cell cellularity observed in the lymph nodes of Foxo1[TKO] mice. To test this hypothesis, we next compared the homing of purified Foxo1-deficient or Foxo1-sufficient $CD4_N$ cells 18 h after their transfer into recipient mice (Fig. S4d). In agreement with Kerdiles et al.[23], we found that the ability of Foxo1-deficient $CD4_N$ cells to home to lymph nodes but not to the spleen was significantly impaired compared with that of Foxo1-sufficient $CD4_N$ cells (Fig. S4e). Thus, part of the decrease of the absolute numbers of naive T cells in the SLOs of Foxo1[TKO] mice could result from a decrease in thymic input due to altered trafficking of recent thymic migrants, which would affect the cellularity of the lymph nodes as early as 4 weeks after birth before generalizing to all SLOs in 12-week-old mice.

We next decided to study the behavior of a cohort of thymic migrants depending on their expression of Foxo1. To do so, we adoptively transferred $CD4_N$ thymocytes from 1-month-old Foxo1[TKO] or Foxo1[Ctrl] CD45.1 mice into CD45.2 mice of the same age and genotype and analyzed their progeny 2 months later (Fig. 2f). Whereas in WT mice, most recovered cells were still naive, the progeny of $CD4_N$ thymocytes lacking the expression of Foxo1 contained high proportions of $CD4_M$ and $CD4_R$ cells (Fig. 2g). More precisely, we recovered at least 20-fold more CD45.1⁺ $CD4_M$, and $CD4_R$ cells in the lymph nodes and spleen of Foxo1[TKO] mice compared to Foxo1[Ctrl] mice (Fig. 2h). Of note, absolute numbers of CD45.1⁺ $CD4_N$ cells in Foxo1[TKO] mice were reduced by 2-fold compared to those in Foxo1[Ctrl] mice. These latter results suggest that in Foxo1[TKO] mice, $CD4_M$, and $CD4_R$ cells deriving from the initially injected naive thymocytes expanded strongly. However, the decreased absolute numbers of CD45.1⁺ $CD4_N$ cells recovered from the periphery of Foxo1[TKO] mice also suggest that injected thymic $CD4_N$ cells converted more efficiently into $CD4_M$ cells and $CD4_R$ cells when lacking Foxo1 expression. So, in Foxo1[TKO] mice, a decrease in thymic input together with a higher differentiation rate of $CD4_N$ cells into $CD4_M$ and $CD4_R$ cells and a higher expansion of these latter cells could lead to a peripheral T-cell compartment resembling that of old WT mice.

## Foxo1 down-regulation with age in memory and regulatory T cells leads to their exhaustion

To find out which other consequences may have Foxo1 down-regulation with age on T-cell physiology, we went back to the transcriptomic signature of $CD4_N$ cells from old mice and analyzed it with the help of the QIAGEN Ingenuity Pathway Analysis (IPA, QIAGEN Inc., https://digitalinsights.qiagen.com/IPA) (Fig. 3a). Eleven canonical pathways were found to be both significant and Z-score positive in $CD4_N$ cells from old mice (numbered from the lowest $p$ value). Far more pathways were significantly enriched in $CD4_N$ cells of young adult Foxo1[TKO] mice compared to their control littermates. Interestingly, 8 out of the 11 canonical pathways defined in old mice were also significantly turned on in $CD4_N$ cells from young adult Foxo1[TKO] mice. Among them, the pathway numbered 2, "T-cell exhaustion signaling pathway", had the lowest $p$ value in both analyses (Fig. 3a). To pursue in that direction, we defined, from the resource published by Crawford et al.[36], a list of genes up- or down-regulated in CD4 T cells undergoing exhaustion in the course of a chronic infection (Day 30 exhaustion signature, Supplementary Data 2). Interestingly, this signature was clearly found in the transcriptional profile of $CD4_N$ cells from either old WT or young adult Foxo1[TKO] mice (Figs. 3b, S5a, b).

We then studied the expression of 3 cell membrane proteins overexpressed by exhausted T cells, namely PD1, TIGIT and CD39, by $CD4_M$, $CD8_M$, and $CD4_R$ cells from our mice of interest and their respective controls. PD1 and TIGIT were clearly overexpressed by peripheral $CD4_M$ and $CD4_R$ cells from old WT or young adult Foxo1[TKO] mice (Figs. 3c, S5c). $CD8_M$ cells from old WT or young adult Foxo1[TKO] mice also exhibited higher PD1 levels than $CD8_M$ cells from young adult WT mice although such an increase was more pronounced in old mice than in young adult Foxo1[TKO] mice (Fig. 3c). TIGIT overexpression was observed only in $CD8_M$ cells from old mice (Fig. S5c). CD39 expression has been shown to define cell exhaustion in tumor-infiltrating CD8 T cells[37]. Accordingly, CD39 expression was up-regulated strongly in peripheral $CD8_M$ cells from both old WT or young adult Foxo1[TKO] mice (Fig. S5d). Most inhibitory receptors are also expressed although to a lesser extent in the course of T-cell activation. To go further, we,

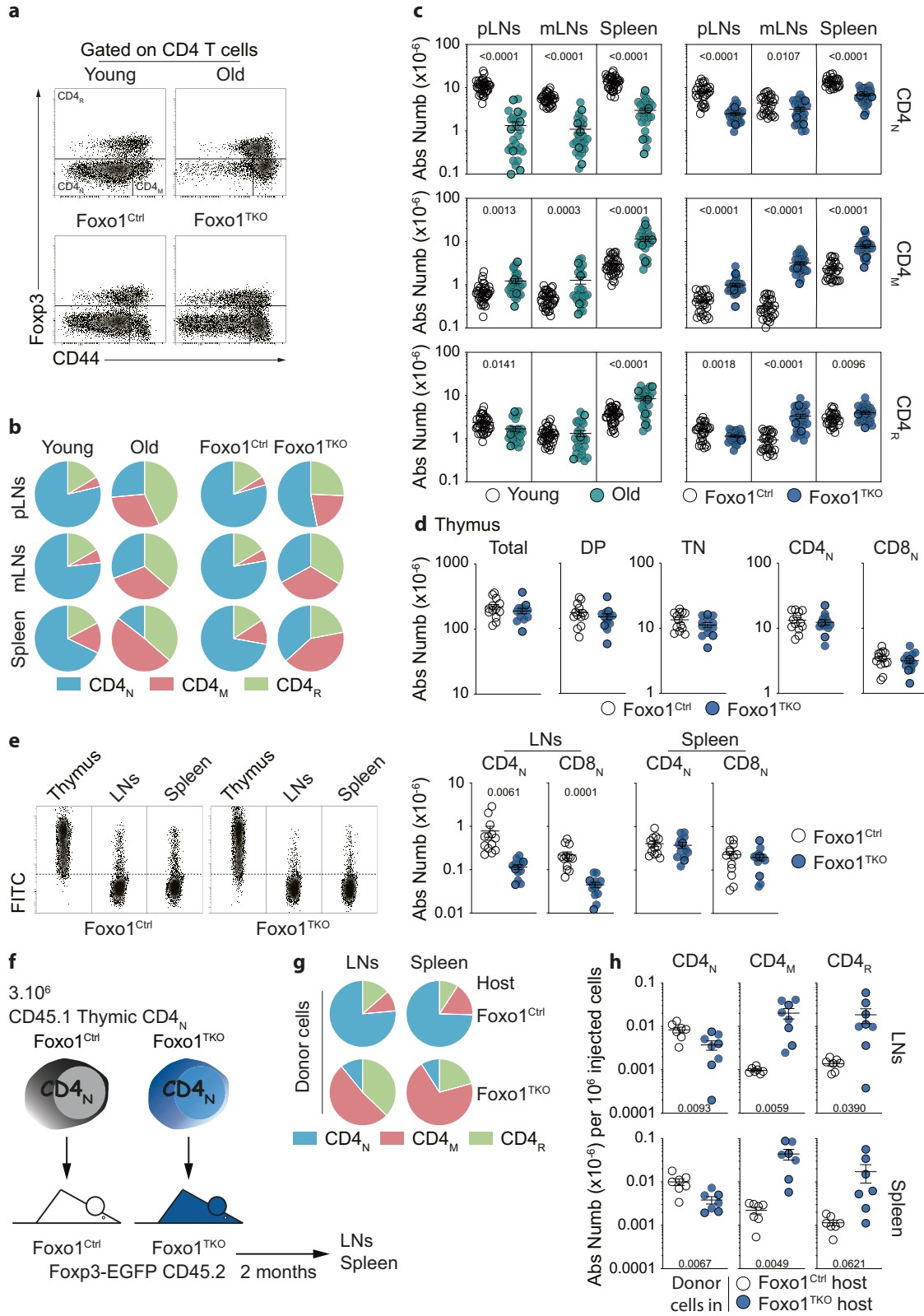

therefore, studied the expression of the transcription factor TOX as it has been shown that TOX is largely dispensable for the generation of effector and memory T cells, but critical for exhaustion[38]. We found that TOX expression was significantly increased in $CD4_M$, $CD4_R$, and $CD8_M$ cells from old WT and young adult $Foxo1^{TKO}$ mice (Fig. S6a). It has been shown that terminally differentiated exhausted T cells express lower levels of the transcription factor TCF-1 than exhausted

T-cell progenitor cells and activated T cells[39–41]. Interestingly, we found that TCF-1 expression was strongly down-regulated in $CD4_M$, $CD4_R$, and $CD8_M$ cells from old WT and young adult $Foxo1^{TKO}$ mice compared to the same cells from control mice (Fig. S6b). Furthermore, these 3 cell subsets contain a significant proportion of $TOX^+$ $TCF\text{-}1^{-/low}$ cells in both old WT and young adult $Foxo1^{TKO}$ mice (Fig. S6c). Altogether, these results strongly suggest that Foxo1 down-regulation with age in

**Fig. 2 | Similar distribution of CD4 T-cell subsets in the SLOs of old WT and young adult Foxo1$^{TKO}$ mice. a** Foxp3/CD44 representative dot-plots are shown for spleen CD4 T cells from the indicated representative mice. **b** Distribution of CD4$_N$, CD4$_M$, and CD4$_R$ cells among CD4 T cells in the indicated SLOs (pLNs (peripheral Lymph Nodes), mLNs (mesenteric Lymph Nodes) and Spleen) of young adult and old WT mice versus young adult Foxo1$^{Ctrl}$ and Foxo1$^{TKO}$ mice. **c** Total cell numbers of CD4$_N$, CD4$_M$ and CD4$_R$ cells recovered from the SLOs of the indicated mice. **d** Absolute cell numbers of total, double-positive (CD4$^+$CD8$^+$), triple-negative (CD4$^-$CD8$^-$ TCR$_\beta^-$), CD4$_N$ and CD8$_N$ thymocytes from 1-month-old Foxo1$^{Ctrl}$ and Foxo1$^{TKO}$ mice. **e** 1-month-old Foxo1$^{Ctrl}$ and Foxo1$^{TKO}$ mice were injected intrathymically with FITC and sacrificed 16 h later. FITC/FSC representative dot-plots are shown for CD4 T cells from the spleen, lymph nodes, and the thymus of representative mice (left panel). Absolute cell numbers of FITC$^+$ CD4$_N$ and CD8$_N$ cells from the spleen and lymph nodes (right panel). **f–h** $3.10^6$ CD4$_N$ cells from the thymus of 1-month-old CD45.1 Foxo1$^{TKO}$ or Foxo1$^{Ctrl}$ mice were injected i.v. into 1-month-old CD45.2 Foxo1$^{TKO}$ or Foxo1$^{Ctrl}$ mice respectively. Diagram illustrating the experimental model (**f**). Distribution of CD4$_N$, CD4$_M$, and CD4$_R$ cells among donor CD4 T cells recovered from the indicated SLOs of recipient mice (**g**). Absolute cell numbers of CD4$_N$, CD4$_M$, and CD4$_R$ donor cells recovered from the indicated SLOs of recipient mice per $10^6$ injected cells (**h**). Quantifications are represented as Means ± SEM. The significance of differences between two series of results was assessed using Student's unpaired *t* test (**c–e**, **h**). Significant ($p < 0.05$) or almost significant ($0.05 < p < 0.10$) *p*-values are indicated. Source data are provided as a Source Data file.

peripheral memory and regulatory T cells participate to their exhaustion. However, the overexpression of exhaustion markers is, in some cases, less marked in T cells from young adult Foxo1$^{TKO}$ mice than in those from old WT, especially for CD8$_M$ cells. Thus, mechanisms other than Foxo1 down-regulation may also be involved in age-related T-cell exhaustion.

To assess whether such an overexpression of inhibitory receptors would alter the proliferation rate of CD4$_M$, CD8$_M$, and CD4$_R$ cells, we then analyzed the expression of KI67, a protein that is constitutively expressed in cycling mammalian cells and is widely used as a cell proliferation marker. In all SLOs of old mice, fewer CD4$_M$, CD8$_M$, and CD4$_R$ cells expressed KI67 (Fig. 3d). Such a decrease was not observed in CD4$_R$ and CD8$_M$ cells from young adult Foxo1$^{TKO}$ mice. Indeed, the decrease was significant only in CD4$_M$ cells in the spleen and mesenteric lymph nodes of Foxo1$^{TKO}$ mice. Other mechanisms not involving the down-regulation of Foxo1 expression could therefore be involved in the age-related decrease in CD4$_R$ and CD8$_M$ cell proliferation.

Interestingly, in old mice, the less CD4$_M$, CD8$_M$ or CD4$_R$ cells express Foxo1, the more they express PD1 (Fig. 3e). These latter results are also suggesting that Foxo1 down-regulation participate to T-cell exhaustion with age. Of note, PD1 expression in T cells from the SLOs of old mice was also clearly inversely correlated with the expression of KI67 (Fig. 3f).

To confirm that Foxo1 expression prevents T cells from the overexpression of exhaustion markers, we purified CD4$_N$ cells from Foxo1$^{TKO}$ mice and Foxo1$^{Ctrl}$ mice and stimulated them in vitro for 2 or 4 days (Fig. 4a). Foxo1-deficient T cells significantly expressed more PD1 and TIGIT upon activation (Fig. 4b, c). Moreover, the proliferation of Foxo1-deficient T cells was markedly reduced (Fig. 4d). Of note, similar results were obtained by comparing the progeny of CD8$_N$ cells from Foxo1$^{TKO}$ mice and Foxo1$^{Ctrl}$ mice 4 days after activation (Fig. S7a–d). To extend these data, we then determined the transcriptome of the progeny of CD4$_N$ cells deficient or proficient for Foxo1 expression after 4 days of activation. This transcriptomic signature was clearly related to the one we defined from Crawford et al.[36]. (Day 8 exhaustion signature, Supplementary Data 3, Figs. 4e, S7e). The transcription of the key inhibitory receptors Pdcd1 (PD1), Pdcd1lg2 (PD-L2), Tigit, Rgs16, Cd200 and Cd200r was up-regulated while the transcription of others was not affected (Lag3 and Havcr2 (Tim-3)) or even significantly decreased (Entpd1 (CD39), Btla and Ctla4) (Fig. 4f)[37,42,43]. The transcription of several transcription factors known to participate to T-cell exhaustion[36,42,44–47] such as Egr3, Fos, Ikzf2 (Helios), 2 members of the Nr4a family, Eomes, Nfatc1 (Nfat2) and Tox was significantly up-regulated in activated T cells lacking Foxo1 expression (Fig. 4f). By contrast, the expression of Batf and Irf4, 2 transcription factors that were recently shown to cooperate to counter T-cell exhaustion[48], was significantly diminished. Altogether, our results suggest that Foxo1 deficiency in CD4$_N$ and CD8$_N$ cells could lead to their exhaustion upon activation.

Hu et al. have sequenced mRNAs from CD4$_N$ cells from young and old healthy donors after 5 days of activation with anti-CD3/anti-CD28 beads[49]. They have then defined a set of genes differentially expressed depending on whether the CD4 T cells were initially recovered from young or aged donors. Interestingly, the transcriptomic signature thus defined (Supplementary Data 4) corresponds perfectly to the set of genes differentially expressed after activation of CD4$_N$ cells deficient or proficient for Foxo1 expression (Figs. 4g, S7f). A decrease in Foxo1 expression in human CD4$_N$ cells with age could explain the overlap between these two transcriptomic signatures.

## Foxo1 down-regulation in T cells with age and their subsequent aging are determined by their environment

We then sought to discover the cause of Foxo1 down-regulation in T cells with age. First, we noticed that, in old mice, the extent of Foxo1 down-regulation in a given T-cell subset correlated with the levels observed in all other T-cell subsets in all SLOs studied (Fig. 5a). Since the T-cell compartment is heterogeneous in many respects, including the age of its component cells, such synchrony suggests that the decrease in Foxo1 expression would result from a response to an extrinsic stimulus affecting all cells rather than to an internal clock. To verify such a hypothesis, we injected purified total CD4 T cells from 2-month-old CD45.2$^+$ mice into 2- or 21-month-old CD45.1/2$^+$ mice. The fate of adoptively transferred cells was studied 4 weeks later (Fig. 5b). When immerged into an old environment, "young" T cells rapidly down-regulated Foxo1 expression. Consequently, transferred T cells (CD4$_N$, CD4$_M$, and CD4$_R$ cells) expressed Foxo1 levels similar to those observed in T cells from recipient mice (Figs. 5c, S8a). Foxo1 down-regulation in T cells from aged mice is thus mediated by cell-extrinsic signals.

Interestingly, the composition of donor-derived CD4 T cells also varies according to the age of recipient mice. The proportions of CD4$_M$ cells were sharply increased in donor cells recovered from the SLOs of old recipient mice (Fig. 5d). Indeed, decreased absolute numbers of donor CD4$_N$ cells and increased numbers of donor CD4$_M$ cells were recovered from the SLOs of old recipient mice compared to their young adult counterparts (Figs. 5e, S8b). Absolute numbers of donor CD4$_R$ cells recovered from the SLOs were not different between the two types of recipient mice. To assess whether differentiation of CD4$_N$ cells into CD4$_M$ cells participate to the differences observed between old and young adult recipient mice, we performed similar experiments but that time by injecting purified CD4$_N$ cells instead of total CD4 T cells (Fig. S8c). In this context, we still found an increase in the proportion and absolute number of CD4$_M$ cells among the CD45.2$^+$ T cells recovered from the SLOs of old recipient mice (Fig. S8d, e). Specifically, the increase in the absolute number of donor-derived CD4$_M$ cells recovered from the spleen of aged versus young adult recipient mice (+49 238 ± 9 592 cells per million cells initially injected) was of the same order of magnitude as the decrease in the absolute number of donor-derived CD4$_N$ cells observed when comparing aged versus young adult recipient mice (−57,001 ± 32 840 cells per million cells initially injected). Thus, as proposed by others, our data support that, in an old environment, conversion of donor CD4$_N$ cells into CD4$_M$ cells may contribute to the decrease in absolute numbers of the former in favor of the latter[50–53].

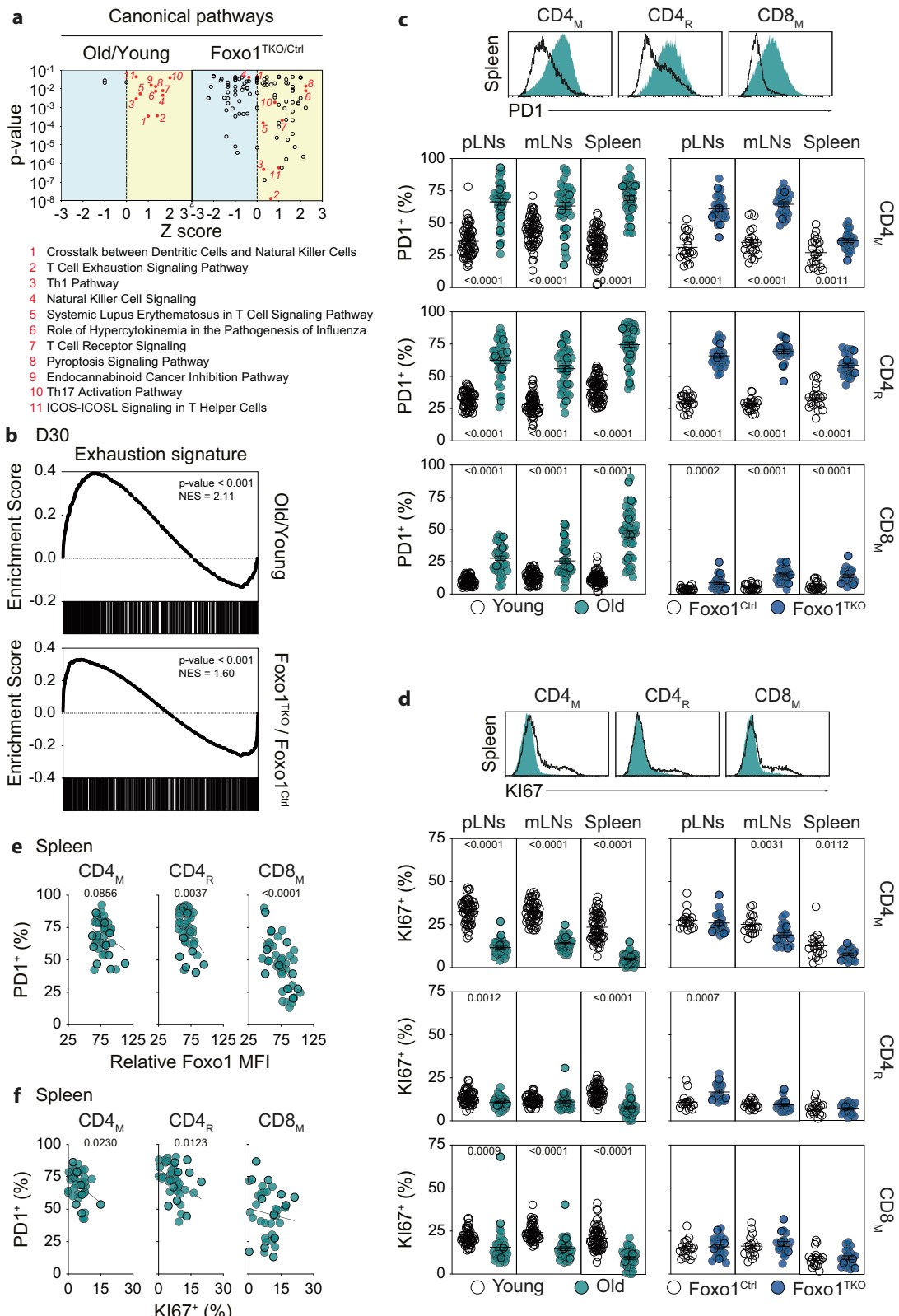

**a** Canonical pathways

1 Crosstalk between Dentritic Cells and Natural Killer Cells
2 T Cell Exhaustion Signaling Pathway
3 Th1 Pathway
4 Natural Killer Cell Signaling
5 Systemic Lupus Erythematosus in T Cell Signaling Pathway
6 Role of Hypercytokinemia in the Pathogenesis of Influenza
7 T Cell Receptor Signaling
8 Pyroptosis Signaling Pathway
9 Endocannabinoid Cancer Inhibition Pathway
10 Th17 Activation Pathway
11 ICOS-ICOSL Signaling in T Helper Cells

**b** D30

**e** Spleen

**f** Spleen

We also studied the phenotype of donor $CD4_M$ and $CD4_R$ cells depending on the age of recipient mice. Increased proportions of cells expressing PD1 and TIGIT were found in the progeny of CD4 T cells 1 month after their transfer into old recipient mice, these proportions reaching the levels observed in the same subsets from recipient mice (Figs. 5f, S8f). Proliferation (as assessed by KI67 expression) of donor-derived $CD4_M$ cells was strongly decreased when transferred into aged recipient mice, compared to their proliferation when transferred into young adult recipient mice, while such decrease was not observed for $CD4_R$ cells (Fig. 5g). Of note, PD1 expression by donor $CD4_R$ and $CD4_M$ cells was inversely correlated with their expression of Foxo1 (Fig. 5h). Finally, the more $CD4_R$ and $CD4_M$ cells expressed PD1, the less they were proliferating (Fig. 5i).

**Fig. 3 | Memory and regulatory T cells from the SLOs of old WT and young adult Foxo1[TKO] mice exhibit an exhausted phenotype. a** $p$ value/Z score dot-plots representing the canonical pathways determined by Ingenuity Pathway Analysis software as significantly ($p < 0.05$) involved in the transcriptomic signature of CD4$_N$ cells from old WT (left panel) or young adult Foxo1[TKO] (right panel) mice. **b** A list of genes up-or down-regulated in CD4 T cells undergoing exhaustion in the course of a chronic infection was determined (GSE30431 Day 30). Expression of these genes by CD4 T$_N$ cells from old WT (upper panel) or young adult Foxo1[TKO] (lower panel) mice was visualized using the GSEA software. **c** PD1 fluorescence histograms of the indicated T-cell subsets from the spleen of a representative old mouse and a representative young adult mouse are shown (upper panel). Percentages of PD1$^+$ cells among CD4$_M$, CD4$_R$, and CD8$_M$ cells are shown for the indicated SLOs of old/ young adult mice (lower left panel) and Foxo1[TKO]/Foxo1[Ctrl] mice (lower right panel). **d** Same as in (**c**) for KI67. **e** Percentages of PD1$^+$ cells among CD4$_M$, CD4$_R$ and CD8$_M$ cells from the spleen of old mice are shown as a function of their Foxo1 relative expression. **f** Percentages of PD1$^+$ cells among CD4$_M$, CD4$_R$ and CD8$_M$ cells from the spleen of old mice are shown as a function of their expression of KI67. Quantifications are represented as Means ± SEM. The significance of differences between two series of results was assessed using Student's unpaired $t$ test (**a, c, d**). The p-values for the GSEA test statistics are calculated by permutation (**b**). For assessing correlations, Pearson correlation (two-sided test, coefficient and 95% confidence intervals) was used (**e, f**). Significant ($p < 0.05$) or almost significant ($0.05 < p < 0.10$) p-values are indicated. Source data are provided as a Source Data file.

---

Overall, our results show that after only 1 month into an old environment, a cohort of transferred "young" CD4 T cells closely resembles old endogenous T cells in terms of decreased Foxo1 expression, enrichment of CD4$_M$ cells at the expense of CD4$_N$ cells, acquisition of an exhausted phenotype and decreased turn-over. Ageing of T cells would be thus mediated by cell-extrinsic signals.

We then injected purified total CD4 T cells from 21-month-old CD45.2$^+$ mice into 2-month-old CD45.1/2$^+$ mice. The fate of adoptively transferred cells was studied 1 month later (Fig. S9a). When immersed in a young environment, old T cells remained old. Indeed, Foxo1 expression did not return to the levels observed in T cells from young adult mice (Fig. S9b). The proportions of CD4$_M$ and CD4$_R$ cells among donor cells were still greatly increased compared with the same proportions observed among cells from recipient mice (Fig. S9c). Finally, PD1 and TIGIT were still overexpressed by donor CD4$_M$ cells, and proliferation of donor-derived CD4$_M$ cells was still greatly diminished compared with endogenous CD4$_M$ cells (Fig. S9d–f). Overall, these results suggest that T-cell aging is not reversible after immersion in a young environment for 1 month.

## Type 1 interferons are involved in T-cell aging

To find out the extrinsic stimulus that mediates Foxo1 down-regulation with age, we went back to Qiagen IPA analysis of the transcriptomic signature of CD4$_N$ cells from old mice. Among identified upstream regulators, we focused on cytokines or cytokine-related genes (Fig. 6a). Interestingly, most Z-scores were positive for cytokines. Among those with a Z-score greater than 2, indicating that these cytokines are highly likely to play a role in the transcriptomic signature of CD4$_N$ cells from old mice, we found mainly pro-inflammatory cytokines (Fig. 6a). Furthermore, when the transcriptome of CD4$_N$ cells was screened for hallmarks predefined by the GSEA program, the 4 most significant signatures found were those of 4 pro-inflammatory cytokines, namely TNF-α, IFN-γ, IL-2, and type 1 interferons (Fig. 6b). Interestingly, type 1 interferons also represent most of the master regulators identified by IPA analysis (Fig. S10a).

It has been shown that baseline levels of pro-inflammatory cytokines increase with age[3,54,55]. We therefore hypothesized that at least one of these cytokines could induce Foxo1 down-regulation. To test this hypothesis, T cells from young adult mice were purified and cultured for 4 days in the presence of one of the indicated cytokines (Fig. 6c). 3 out of the 4 cytokines identified by the GSEA analysis, namely IFN-α, IL-2, and TNF-α, induced significant decrease in Foxo1 expression in CD4$_N$ cells (Fig. 6d). Interestingly, while IFN-α, IL-2 and TNF-α induced Foxo1 down-regulation, IL-6 induced an increase of its expression (Fig. 6d). The effect of several cytokines on Foxo1 expression varied depending on the T-cell subset with, for example, IL-12 increasing Foxo1 expression in CD8 T-cell subsets only (Figs. 6d, S10b). Interestingly, only IFN-α induced significant Foxo1 down-regulation in all T-cell subsets.

The ImmGen consortium has determined the transcriptomic signature of CD4 T cells from the spleen of C57BL/6 mice 2 h after injection of 10, 000 Units of IFN-α (GSE75202 and GSE124829, Fig. 6e).

From GSE75202[56], we defined the transcriptomic signature of CD4 T cells in response to type 1 interferons in vivo (Supplementary Data 5). We found that this signature closely correlated with the signature of CD4$_N$ cells induced by aging (Figs. 6f, S10c). From these datasets, we then determined the relative expression of Foxo1 and Foxo3 in spleen CD4 T cells depending on whether the mice were injected 2 h earlier with IFN-α (Fig. 6g). Surprisingly, we observed that a 2-h exposure to IFN-α in vivo was sufficient to induce significant down-regulation of Foxo1, but not Foxo3, mRNAs (Fig. 6g). Type 1 interferons are thus capable to induce Foxo1 down-regulation in vitro and in vivo to a similar extent to that observed in T cells from old mice (Fig. 1). Accordingly, the transcriptional signature of CD4$_N$ cells from Foxo1[TKO] mice correlate quite well with the signature of CD4 T cells in vivo exposed to type 1 interferon (Fig. 6h). Finally, from GSE46525 public data[26,57], we determined a list of genes for which Foxo1 binds to the promoter region (Supplementary Data 6). Interestingly, the transcription level of nearly all of these genes was strongly decreased in spleen CD4 T cells 2 h after type 1 interferon injection (Fig. 6i). These analyses reinforce the idea that type 1 interferons imprint the transcriptional signature of CD4 T cells, in particular by decreasing Foxo1 expression.

We then sought to determine the signaling pathway downstream of the type I IFN receptor leading to Foxo1 down-regulation. To do so, we cultured purified T cells in the presence of IFN-α alone or with several inhibitors of key kinases for 4 days. (Fig. 7a). As expected, inhibitors of TYK2 and JAK1, 2 kinases directly associated with the 2 chains of the type 1 interferon receptor completely abolished (CD4 T-cell subsets) or reduced (CD8 T-cell subsets) type 1 interferon-induced down-regulation of Foxo1 (Figs. 7b, S10d). Surprisingly, STAT1 inhibition had no effect suggesting that the canonical pathway involving STAT proteins did not appear to be required to induce Foxo1 down-regulation. In fact, as demonstrated by the use of PI3K and AKT inhibitors, type 1 interferon-induced down-regulation of Foxo1 involves the PI3K/AKT pathway in this setting (Figs. 7b, S10d). Indeed, in the presence of inhibitors of PI3K and AKT, Foxo1 down-regulation in response to IFN-α was either completely abolished (in all T-cell subsets for the inhibitor of PI3K and in CD4 T cells for the inhibitor of AKT) or at least significantly reduced (in CD8 T cells for the inhibitor of AKT).

Finally, we assessed whether pre-exposure to IFN-α was sufficient to modulate the response of CD4$_N$ cells upon activation. To do so, we first incubated CD4$_N$ cells from WT mice in vitro with or without IFN-α for 4 days and then stimulated them for 4 days (Fig. 7c). After a 4-day preincubation with IFN-α, T cells significantly expressed more PD1 upon activation (Fig. 7d) and their proliferation was significantly reduced (Fig. 7e), a response quite similar to that of CD4$_N$ cells from Foxo1[TKO] mice (Fig. 4a–d).

To confirm the involvement of type 1 interferons on the aging of the T-cell compartment in vivo, we then compared 22-month-old WT mice to 22-month-old mice deficient for the expression of the type 1 interferon receptor (mice knock-out for the subunit 1 of the type I IFN receptor, Ifnar[KO] mice[58],). We found that all T-cell subsets expressed

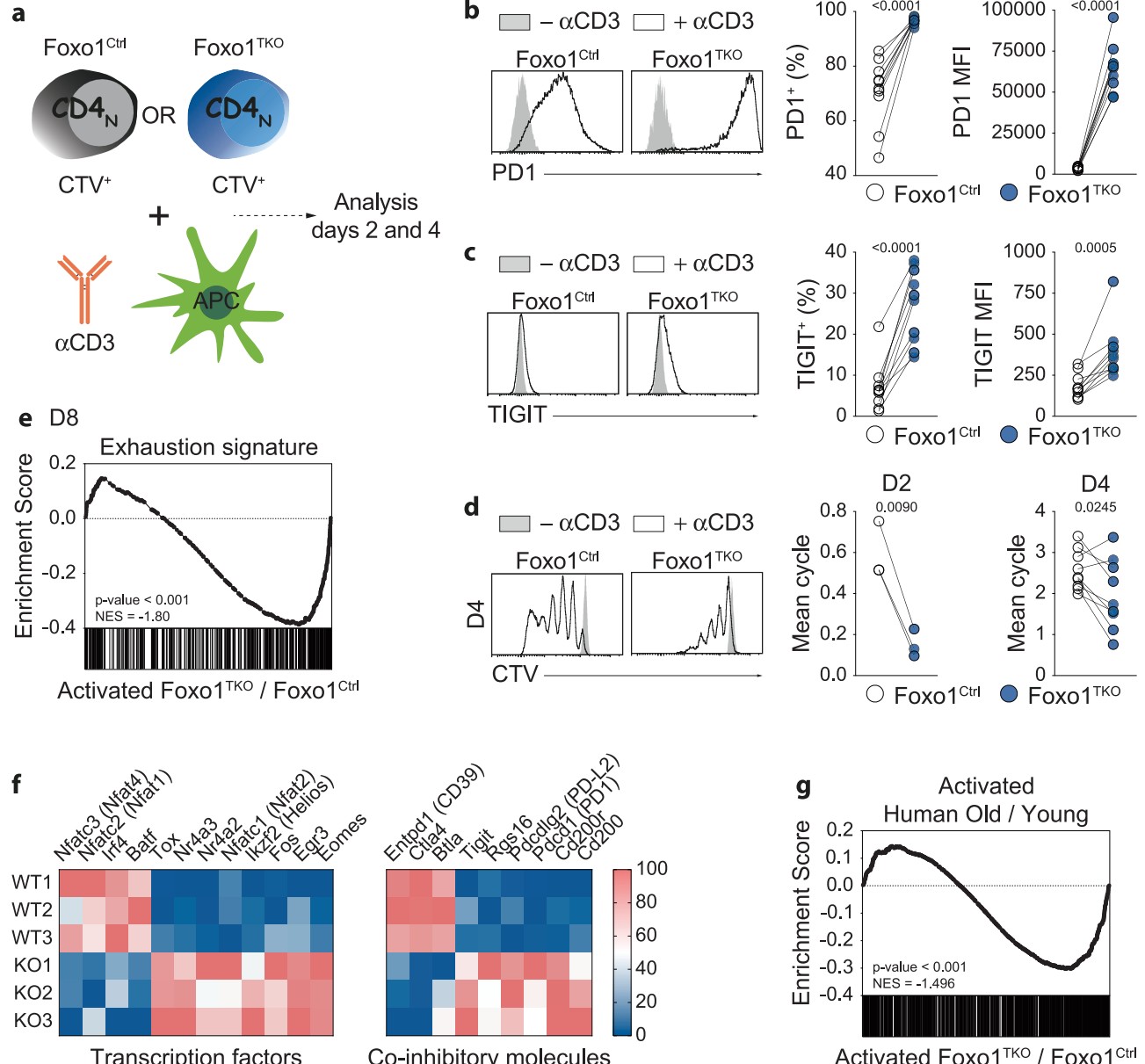

**Fig. 4 | Foxo1 deficiency leads to T-cell exhaustion upon activation. a−d** CTV-labeled purified CD4$_N$ cells from Foxo1$^{TKO}$ and Foxo1$^{Ctrl}$ mice were stimulated for 2 or 4 days with anti-CD3 in the presence of splenocytes from CD3$_\varepsilon^{KO}$ mice. Diagram illustrating the experimental model (**a**). PD1 (**b**) and TIGIT (**c**) fluorescence histograms of T cells after 4 days of culture are shown for a representative experiment (left panel). Percentages of PD1$^+$ cells and PD1 MFI (**b**, $n = 10$ independent experiments) and percentages of TIGIT$^+$ cells and TIGIT MFI (**c**, $n = 10$ independent experiments) among the progeny of CD4$_N$ cells after 4 days of culture (right panel). CTV fluorescence histograms of T cells after 4 days of culture are shown for a representative experiment (**d**, left panels). The average number of cell cycles was calculated and plotted (**d**, right panel, $n = 3$ independent experiments for day 2, $n = 9$ independent experiments for day 4). Each pair of dots represents an individual experiment. **e** Expression of genes characterizing exhausted CD4 T cells 8 days

after LCMV infection (GSE30431 Day 8) by the progeny of activated CD4$_N$ cells from Foxo1$^{TKO}$ versus Foxo1$^{Ctrl}$ mice after 4 days of culture was visualized using the GSEA software. **f** Expression pattern of chosen genes differentially expressed by exhausted CD4 T cells (±1.5 fold change, with a $p$ value of <0.05) by the progeny of activated CD4$_N$ cells from Foxo1$^{TKO}$ versus Foxo1$^{Ctrl}$ mice after 4 days of culture. **g** Expression of genes characterizing the progeny of activated CD4$_N$ cells from old versus young healthy donors 5 days after activation (SRP158502) by the progeny of CD4$_N$ cells from Foxo1$^{TKO}$ versus Foxo1$^{Ctrl}$ mice after 4 days of culture was visualized using the GSEA software. The significance of differences between two series of results was assessed using Student's paired $t$ test (**b**−**d**). The p-values for the GSEA test statistics are calculated by permutation (**e**, **g**). Significant ($p < 0.05$) p-values are indicated. Source data are provided as a Source Data file.

higher levels of Foxo1 when recovered from the SLOs of old Ifnar$^{KO}$ mice than when recovered from old WT mice (Figs. 8a, S11a). Accordingly, CD4$_N$ and CD8$_N$ cells recovered from the spleen of old Ifnar$^{KO}$ mice expressed higher levels of CCR7 and CD62L than when recovered from the spleen of old WT mice (Fig. S11b) confirming that the expression of these two proteins was closely linked to that of Foxo1 in naive T cells (Fig. S2e−j, Figs. 8b, S11b). However, in old Ifnar$^{KO}$ mice, although increased compared with old WT mice, Foxo1 expression in

T cells is still decreased compared with young adult WT mice. Thus, other extrinsic factors, especially other inflammatory cytokines, may also participate in the decrease of Foxo1 expression with age.

The composition of the peripheral T-cell compartment in terms of proportions of naive, memory and regulatory T cells varied according to the ability of mice to express the type 1 interferon receptor. Indeed, proportions of CD4$_M$, CD4$_R$, and CD8$_M$ cells were decreased in favor of CD4$_N$ and CD8$_N$ cells in the SLOs of old Ifnar$^{KO}$ mice when compared to

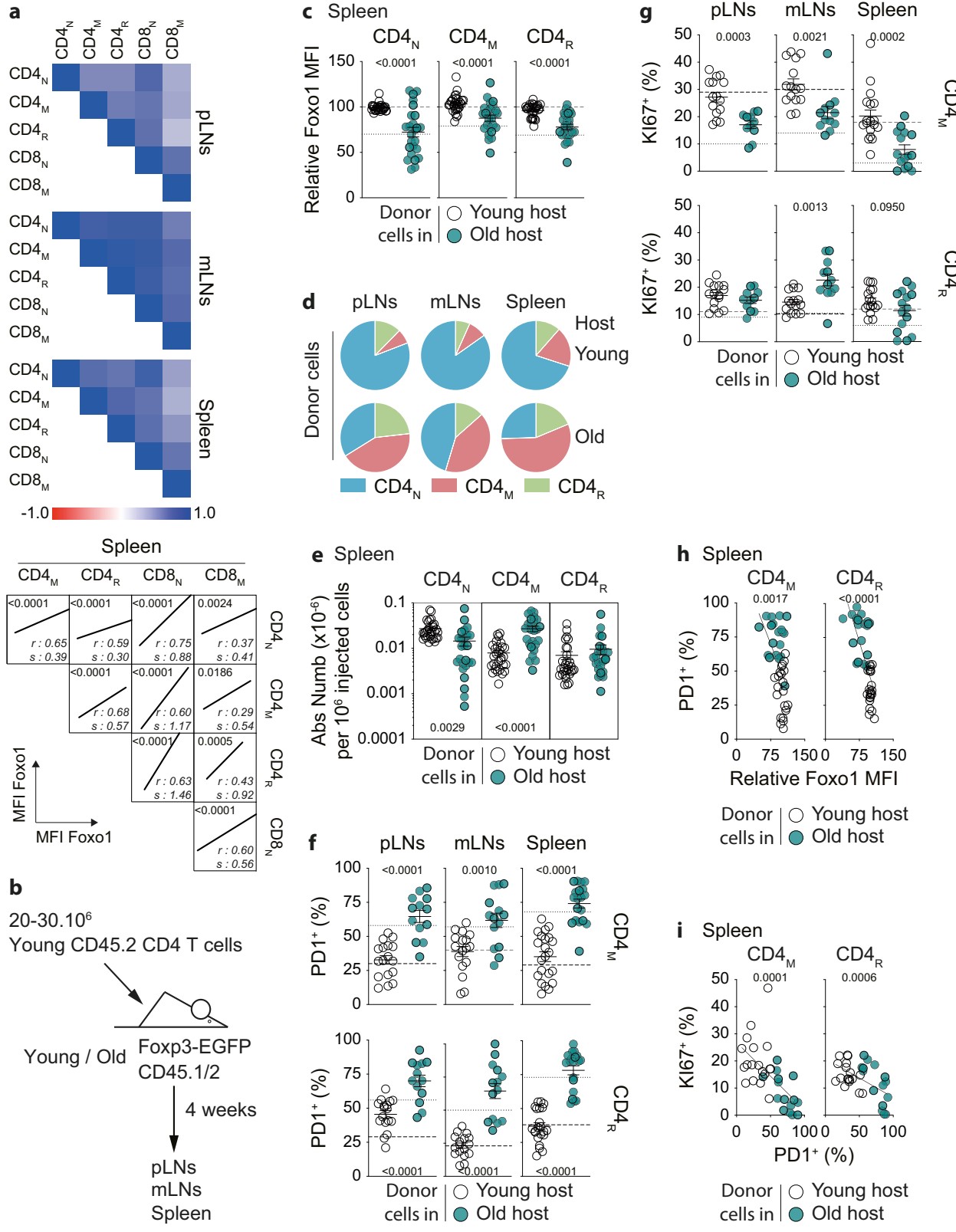

the SLOs of old WT mice (Fig. 8c). PD1 expression by $CD4_M$ cells was significantly decreased in the SLOs of old Ifnar[KO] mice compared with WT mice whereas such a decrease was observed only in mesenteric lymph-nodes for $CD8_M$ cells and in peripheral lymph-nodes for $CD4_R$ cells (Fig. 8d). These results suggest that type 1 interferons may be key actors in inducing age-related PD1 expression in $CD4_M$ cells but may be less involved in age-related PD1 expression in $CD4_R$ and $CD8_M$ cells.

However, when 18-month-old WT and Ifnar[KO] mice were compared, PD1 expression in $CD4_M$, $CD4_R$, and $CD8_M$ cells was significantly lower in all SLOs of Ifnar[KO] mice compared to PD1 expression by their T-cell counterparts from WT mice (Fig. S11c). These latter results suggest that type 1 interferons may participate in the overexpression of PD1 by all T-cell subsets with age, but that other factors may compensate for the lack of expression of the type 1 interferon receptor in $CD4_R$ and $CD8_M$

**Fig. 5 | Cell-extrinsic signals induce T-cell aging. a** Correlogram showing for the indicated SLO of old mice the correlations between Foxo1 expression by a given subset of T cells and its expression by the others (upper panel). Blue colors indicate positive correlations, while red colors indicate negative correlations. The darker the color, the more significant the correlation. Correlation curves are presented for the spleen (lower panel). **b** Diagram illustrating the experimental model. **c** Relative Foxo1 MFIs for CD4$_N$, CD4$_M$ and CD4$_R$ donor cells recovered from the spleen of recipient mice are shown. The dashed lines represent the average Foxo1 MFI of the corresponding T-cell subset from the spleen of young adult (----) and old (....) recipient mice. **d** Distribution of CD4$_N$, CD4$_M$ and CD4$_R$ cells among donor CD4 T cells recovered from the indicated SLOs of young adult versus old recipient mice. **e** Absolute cell numbers of CD4$_N$, CD4$_M$ and CD4$_R$ donor cells recovered from the spleen of young adult versus old recipient mice per $10^6$ injected cells. Percentages of PD1$^+$ (**f**) and KI67$^+$ (**g**) cells among CD4$_M$ and CD4$_R$ donor cells are shown for the indicated SLOs of old versus young adult recipient mice. The dashed lines represent the average percentages for the corresponding T-cell subset of young adult (----) and old (....) recipient mice. **h** Percentages of PD1$^+$ cells among CD4$_M$ and CD4$_R$ donor cells recovered from the spleen of recipient mice are shown as a function of their Foxo1 relative expression. **i** Percentages of KI67$^+$ cells among CD4$_M$ and CD4$_R$ donor cells recovered from the spleen of recipient mice are shown as a function of their expression of PD1. Quantifications are represented as Means ± SEM. The significance of differences between two series of results was assessed using Student's unpaired $t$ test (**c**, **e**, **f**, **g**). For assessing correlations, Pearson correlation (two-sided test, coefficient, and 95% confidence intervals) was used (**a**, **h**, **i**). Significant ($p < 0.05$) or almost significant ($0.05 < p < 0.10$) $p$-values are indicated. Source data are provided as a Source Data file.

cells in very old animals. TIGIT expression by T cells was not affected whatever the ability of old mice to express the type 1 interferon receptor (Fig. S11d). Finally, when compared to WT mice, we observed in old Ifnar$^{KO}$ mice a slight increase in KI67 expression by CD4$_M$, CD4$_R$, and CD8$_M$ cells that reached significance in some SLOs but not in others (Fig. 8e). Overall, with regard to some but not all of the parameters studied, the T-cell compartment of mice lacking expression of the type 1 interferon receptor appears to be less affected by aging than the T-cell compartment of WT mice.

## Discussion

Delpoux et al. have recently shown that Foxo1 expression was down-regulated with age in human CD8 T cells[59]. Here, we show that Foxo1 expression is also down-regulated in T cells from old mice when compared to young adult mice, that this decrease is gradual and results from a decrease in Foxo1 transcription in T cells (Figs. 1, S1). Our results also strongly suggest that age-related regulation of Foxo1 expression may be induced by an age-related increase in inflammatory cytokine levels, particularly type 1 interferons. Moreover, we have shown that, in vitro, type 1 interferons induce Foxo1 down-regulation and that this involves the PI3K/AKT signaling pathway (Fig. 7). Interestingly, AKT phosphorylation is increased ex vivo in naive T cells from old mice, and the extent of AKT phosphorylation correlates with the extent of Foxo1 down-regulation in these cells (Figs. 1, S2). Thus, taken together, our data suggest that the progressive decrease in Foxo1 expression with age may first result from chronic activation of the PI3K/AKT signaling pathway induced by inflammatory cytokines at least in CD4$_N$ and CD8$_N$ cells. This question remains open for the other T-cell subsets as we did not observe a clear correlation between AKT phosphorylation and Foxo1 levels in their cases. Secondly, as the transcription factor Foxo1 increases its own transcription[60,61], a decrease in Foxo1 expression at the protein level should ultimately lead to a decrease in Foxo1 transcription.

Importantly, the decrease in Foxo1 expression in T cells with age imprints their transcriptional signature. Indeed, we found that the transcriptional profile of CD4$_N$ cells from aged mice was very similar to that defined from CD4$_N$ cells from young adult mice lacking Foxo1 expression in T cells (Figs. 1h, S2d). Such changes in the mRNA content of T cells with age may have consequences for their behavior. In particular, it has long been known that Foxo1 governs the expression of many genes involved in T-cell trafficking such as CD62L, CCR7 and S1PR1[22,23]. Thus, the decrease in Foxo1 expression with age could have an impact on their ability to circulate between lymphoid and non-lymphoid tissues. Interestingly, we recently showed that the percentage of resident T cells in SLOs increases significantly with age, with S1PR1 down-regulation possibly contributing to this altered homeostasis[30]. Of note, resident memory T cells also accumulate with age in non-lymphoid tissues such as fat[62], skin[63], and lungs[64].

Surprisingly, we found that the transcriptional signature of CD4$_N$ cells from old mice has some similarities to the signature that defines exhausted effector CD4 T cells[36] (Fig. 3). Gustafson et al. found that miRNA expression profiles of human CD8$_N$ cells from elderly individuals exhibit miRNA expression profiles partially related to cell differentiation[65]. Furthermore, Eberlein et al. and Pulko et al. showed that the transcriptional signatures of mouse and human CD8$_N$ and CD8$_M$ cells gain similarity with age[66,67]. Finally, when studying chromatin accessibility by ATAC-seq, Moskowitz et al. also found that aged human CD8$_N$ cells were more closely related to CD8$_M$ than CD8$_N$ cells from young individuals[68]. Thus, as suggested by others[9,65–69], our data may reinforce the idea that naive T cells may undergo partial differentiation during aging. Interestingly, the transcriptomic profile of CD4$_N$ cells from young adult Foxo1$^{TKO}$ mice correlates closely with that of CD4$_N$ cells from aged WT mice (Figs. 1, S2), suggesting that Foxo1 down-regulation may be one of the main causes explaining the observed changes in mRNA content of naive T cells with age. Upon adoptive transfer into recipient mice, "young" CD4$_N$ cells give rise to CD4$_M$ cells and the extent of this differentiation process depends on the age of the recipient mice (Fig. S8). Specifically, the increase in the number of CD4$_M$ cells recovered from old recipient mice when compared to young ones is of the same magnitude as the number of CD4$_N$ cells that have disappeared in old recipient mice when compared to what can be recovered from young adult recipient mice. This latter result suggests a greater propensity of CD4$_N$ cells to differentiate into CD4$_M$ cells when immersed in an aged environment. Over the past 20 years, several observations have concordantly indicated that naive T-cell homeostasis is profoundly affected by aging with in particular an increased potential of naive T cells to convert into virtual memory cells[8,69]. Foxo1 is known to be crucial in maintaining cell quiescence and thus oppose proliferation[14,15]. Again, down-regulation of Foxo1 with age may help explain the greater propensity of naive T cells in aged mice to differentiate into memory cells.

T-cell exhaustion has been widely described in three main situations: chronic infections, anti-tumor responses, and aging[8,36,42,70]. During viral and bacterial infections, Foxo1 is essential for promoting differentiation of long-lived memory T cells and thus enhancing recall responses[71–73]. Specifically, Foxo1 is thought to antagonize the differentiation program of naive T cells into short-lived, terminally differentiated effector CD8 T cells to the benefit of generating long-lived memory T cells[57]. Furthermore, Delpoux et al. showed that during acute infections by the mouse cytomegalovirus or the lymphocytic choriomeningitis virus (LCMV) in mice, Foxo1 is required to prevent virus-specific CD8 T-cell anergy and senescence[59,74]. In this paper, we show that memory and regulatory T cells from aged WT mice and young adult Foxo1$^{TKO}$ mice exhibit increased expression of co-inhibitory receptors compared to their cell-counterparts from young adult WT mice (Figs. 3, S5, S6). Furthermore, when stimulated, CD4$_N$ and CD8$_N$ cells from Foxo1$^{TKO}$ mice rapidly express co-inhibitory receptors and transcription factors involved in T-cell exhaustion, and proliferate less efficiently than their cell-counterparts from control mice (Figs. 4, S7). Taken together, these results strongly suggest that Foxo1 could play an important role in preventing T-cell exhaustion. Down-regulation of Foxo1 in T cells may

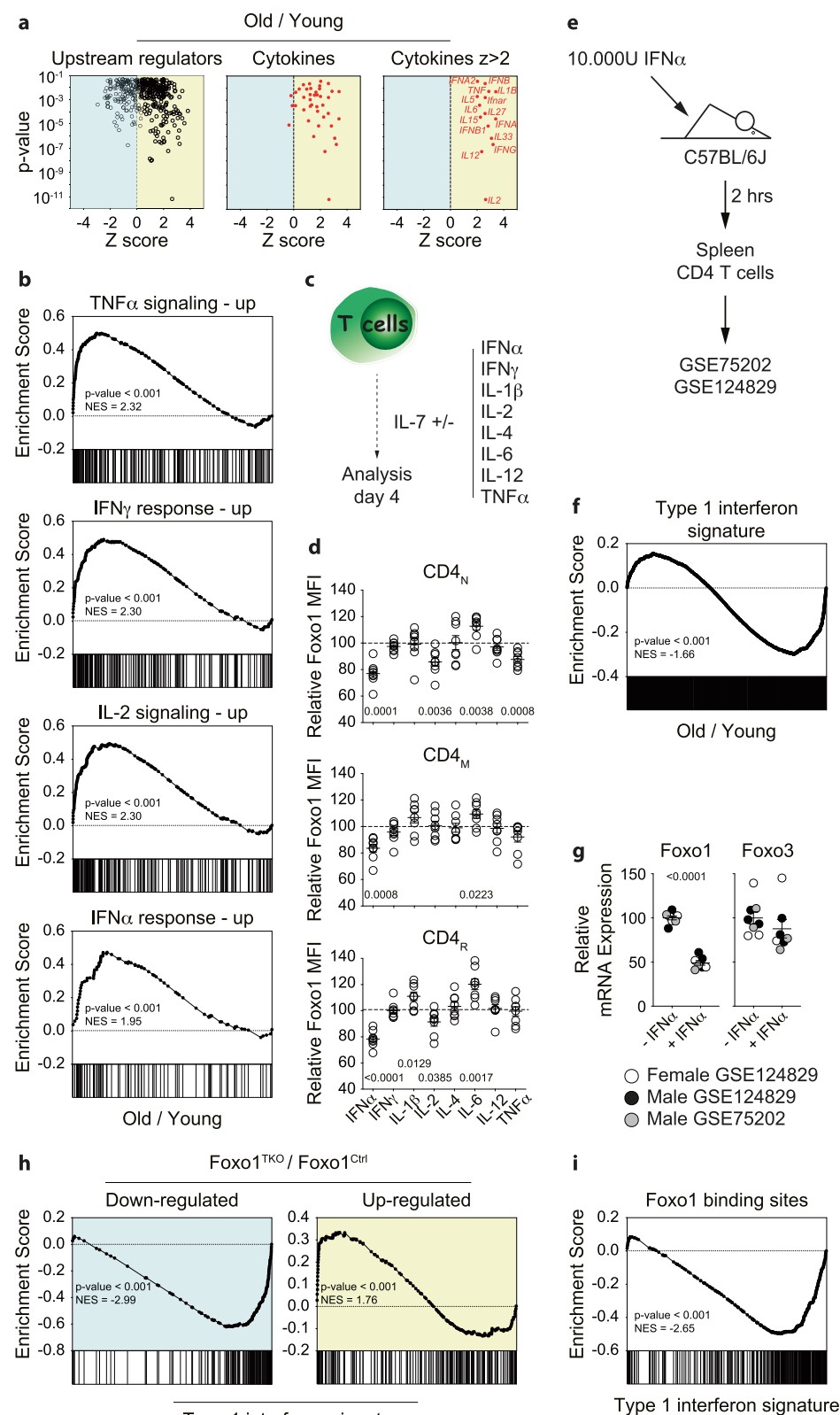

therefore play an important role in their exhaustion and decreased functionality with age.

Our data may help understanding why mice deficient for Foxo1 expression in all T cells (CD4$^{cre}$ Foxo1$^{fl/fl}$ mice alias Foxo1$^{TKO}$ mice) and mice deficient for Foxo1 expression in CD4$_R$ cells only (Foxp3$^{cre}$ Foxo1$^{fl/fl}$ mice) do not develop a similar pathology. Indeed, whereas mice deficient for Foxo1 expression in T cells develop mild mononuclear cell infiltration in non-lymphoid organs such as the heart, salivary glands, kidney, and liver at 1 year of age[24,25], mice deficient for Foxo1 expression only in regulatory T cells show heavy infiltrates in salivary glands, lungs, liver, pancreas, stomach, and colon as early as 21 days after birth and are moribund within 35 days after birth[75]. Foxo1 has been shown to be important for optimal CD4$_R$ cell suppressive capabilities[75,76], and the fulminant phenotype of Foxp3$^{cre}$ Foxo1$^{fl/fl}$ mice has been considered to

**Fig. 6 | Inflammatory cytokines modulate Foxo1 expression in T cells. a** p value/ Z score dot-plots representing all Upstream Regulators (left panel), the cytokines and cytokines-related regulators (middle panel) and among these ones, those with a Z score >2 (right panel) determined by Ingenuity Pathway Analysis software as significantly ($p < 0.05$) involved in the transcriptomic signature of CD4$_N$ cells from old WT mice. **b** Representation of the 4 most significant GSEA Hallmarks gene sets retrieved in the transcriptomic signature of CD4$_N$ cells from old WT mice. **c** Purified T cells from young WT mice were cultured for 4 days with IL-7 alone or together with the indicated cytokines. Diagram illustrating the experimental model. **d** Relative Foxo1 MFIs for CD4$_N$, CD4$_M$ and CD4$_R$ cells recovered after 4 days of culture with IL-7 alone or together with the indicated cytokines. Relative MFIs were calculated after barcoding by dividing the MFI of a given T-cell subset in the presence of one given cytokine by the MFI of the same T-cell subset cultured with IL-7 alone. **e** Diagram illustrating the experimental model depicted in GSE75202 and GSE124829 from the Immunological Genome Project Consortium. **f** A list of genes

up- or down-regulated in CD4 T cells by type 1 interferons was determined from GSE75202 (Supplementary Data 5). Expression of these genes by CD4$_N$ cells from old WT mice was visualized using the GSEA software. **g** Relative Foxo1 and Foxo3 expressions from GSE75202 and GSE124829 were determined by dividing for each mouse the corresponding counts by the average number of counts in the control group. **h** GSEA analysis visualizing the expression of genes down- or up-regulated in Foxo1-deficient CD4$_N$ cells (Supplementary Data 1) by CD4 T cells from the spleen of mice treated as illustrated in (**e**). **i** A list of genes for which Foxo1 binds in the promoter region was determined from GSE46525 (Supplementary Data 6). GSEA analysis visualizing the expression of these genes by CD4 T cells from the spleen of mice treated as illustrated in (**e**). Quantifications are represented as Means ± SEM. The significance of differences between two series of results was assessed using Student's unpaired $t$ test (**a, d, g**). The $p$-values for the GSEA test statistics are calculated by permutation (**b, f, h, i**). Significant ($p < 0.05$) $p$-values are indicated. Source data are provided as a Source Data file.

recapitulate that observed in mice with the Scurfy mutation in the Foxp3 gene or in mice depleted of Treg cells[77,78]. The remaining question is why Foxo1[TKO] mice develop only mild infiltration and this relatively late in life. Our results showing that conventional T cells lacking Foxo1 expression would be prone to exhaustion provide an explanation for this apparent paradox. Indeed, the conventional T-cell compartment of Foxp3[cre] Foxo1[fl/fl] mice is composed of fully functional T cells whose autoreactivity would be unleashed due to defective CD4$_R$ cells, whereas in Foxo1[TKO] mice, conventional T cells, upon activation, would be rapidly exhausted, thus restoring T-cell tolerance to Self.

Our data clearly demonstrate that T-cell extrinsic signals contribute significantly to quantitative and qualitative changes in the peripheral T-cell compartment with age (Fig. 5). These results are in agreement with those of Jergovic et al. who show a role for the priming environment in the defective transcriptional programming of effector CD8 T cells in response to Listeria monocytogenes in aged mice[79]. Transcriptional signature analysis of CD4$_N$ cells from aged mice revealed that they respond to the presence of inflammatory cytokines (Fig. 6). It has been described that levels of inflammatory mediators in the circulation increase with age[3,54,55,80]. In the present paper, we show that type 1 interferons induce Foxo1 down-regulation in vitro and in vivo (Fig. 6) and that the T-cell compartment of old mice deficient for type 1 interferon receptor expression appears less affected by aging with notably increased Foxo1 expression by T cells, higher proportions of naive T cells, and decreased PD1 expression by memory T cells compared to their counterparts from old WT mice (Figs. 8, S11). Type 1 interferons have been shown to play a dual role in chronic viral infection and cancer immunity, with prolonged type 1 interferon signaling potentially leading to immune dysfunction[81,82]. First, type 1 interferon blockade after establishment of chronic LCMV infection leads to enhanced CD4 T-cell and IFN-γ-dependent clearance of the virus[83,84]. In the same line, Cheng et al. have shown that in the HIV-1–infected humanized mice, low levels of type 1 interferon signaling contribute to HIV-1–associated immune dysfunction[85]. Second, in cancer, sustained type 1 interferon signaling can induce immune escape, promote distant metastasis, and contribute to resistance to anti-PD1 immunotherapy[86,87]. Finally, consistent with our observations, type 1 interferons have been shown to induce PD-L1 expression on tumor cells and PD1 expression on immune cells in head and neck squamous cell carcinoma[88] and to synergize with coated anti-CD3/28 antibodies in vitro to induce expression of co-inhibitory receptors such as Tim-3, Lag3, and PD1[89]. Thus, in line with the literature on cancer immunity and chronic infections, our results suggest strongly that sustained type 1 interferon signaling participates in the exhaustion and aging of the T-cell compartment. Furthermore, our data suggest that at least part of the negative effects of exposure to type 1 interferons on immune responses may derive from their ability to induce down-regulation of Foxo1 in T cells. However, only CD4$_M$ cells from 22-month-old Ifnar[KO] mice showed a significant decrease in the expression

of the key inhibitory receptor PD1. Such a decrease is observed in CD4$_R$ and CD8$_M$ cells only in younger animals (18-month-old mice). Thus, our results do not exclude the possibility that additional factors, in particular other inflammatory cytokines, may also contribute to T-cell exhaustion with age. It remains to identify these factors and determine whether or not they act by inducing Foxo1 down-regulation.

Both the type 1 interferon receptor and the transcription factor Foxo1 are expressed by most cell types in mammals. Therefore, sustained type 1 interferon signaling with age can induce Foxo1 down-regulation in many other cell types besides T cells. Interestingly, Foxo1 down-regulation with age has been documented in brain[90], muscle[91], articular cartilage[92,93], intervertebral disks[94] and meniscus[95]. In most cases, the extent of its down-regulation has been correlated with pathologies such as neuronal or intervertebral disc degeneration, osteoarthritis or muscle regenerative failure. Foxo1 down-regulation with age could also explain the development of certain cancers with age such as classical Hodgkin lymphoma[96]. Thus, the type 1 interferon-Foxo1 axis may be involved in aging in mammals well beyond the immune system.

## Methods

### Mice
3-month-old and 22-month-old mice were used for experiments unless otherwise indicated. C57BL/6 Foxp3-GFP CD45.1/2 or CD45.2 mice[97,98], C57BL/6 IfnarKO mice[58,99], C57BL/6 CD3ε[KO 100] and C57BL/6 Foxp3-GFP Foxo1[TKO] and Foxo1[Ctrl] CD45.1 or CD45.2 mice[23,27] were maintained in our own animal facilities, under specific pathogen–free (SPF) conditions. The old mice and their young adult controls were all female mice. Mice were housed in separate individually ventilated cages in the same controlled environment (12h-light-dark cycle, 22 ± 2∘C, 30–70% humidity), with standard rodent chow and water ad libitum. All tests for rodent pathogens carried out during the 5 years we have been working on our aging colony have been negative. All procedures were approved by the ethics committee for animal experimentation n°34 and validated by the "Ministère de l'Enseignement Supérieur, de la Recherche et de l'Innovation" with the number APAFIS ≠ 20630-2018033016303981v5. Sample sizes were chosen to ensure reproducibility of the experiments and in accordance with the 3R of animal ethics regulation.

### Cell suspensions
All mice were sacrificed by vertebral dislocation and peripheral lymph-nodes (pooled cervical, axillary, brachial, and inguinal lymph-nodes; pLNs), mesenteric LNs (mLNs), spleen, and thymus were harvested and homogenized, and passed through a nylon cell strainer (BD Falcon) in RPMI 1640 GlutaMAX (Life Technologies) supplemented with 10% FCS (Biochrom) for adoptive transfer and cell culture (LNs only) or in 5% FCS, 0.1% NaN3 (Interchim) in phosphate buffered saline (PBS) for flow cytometry.

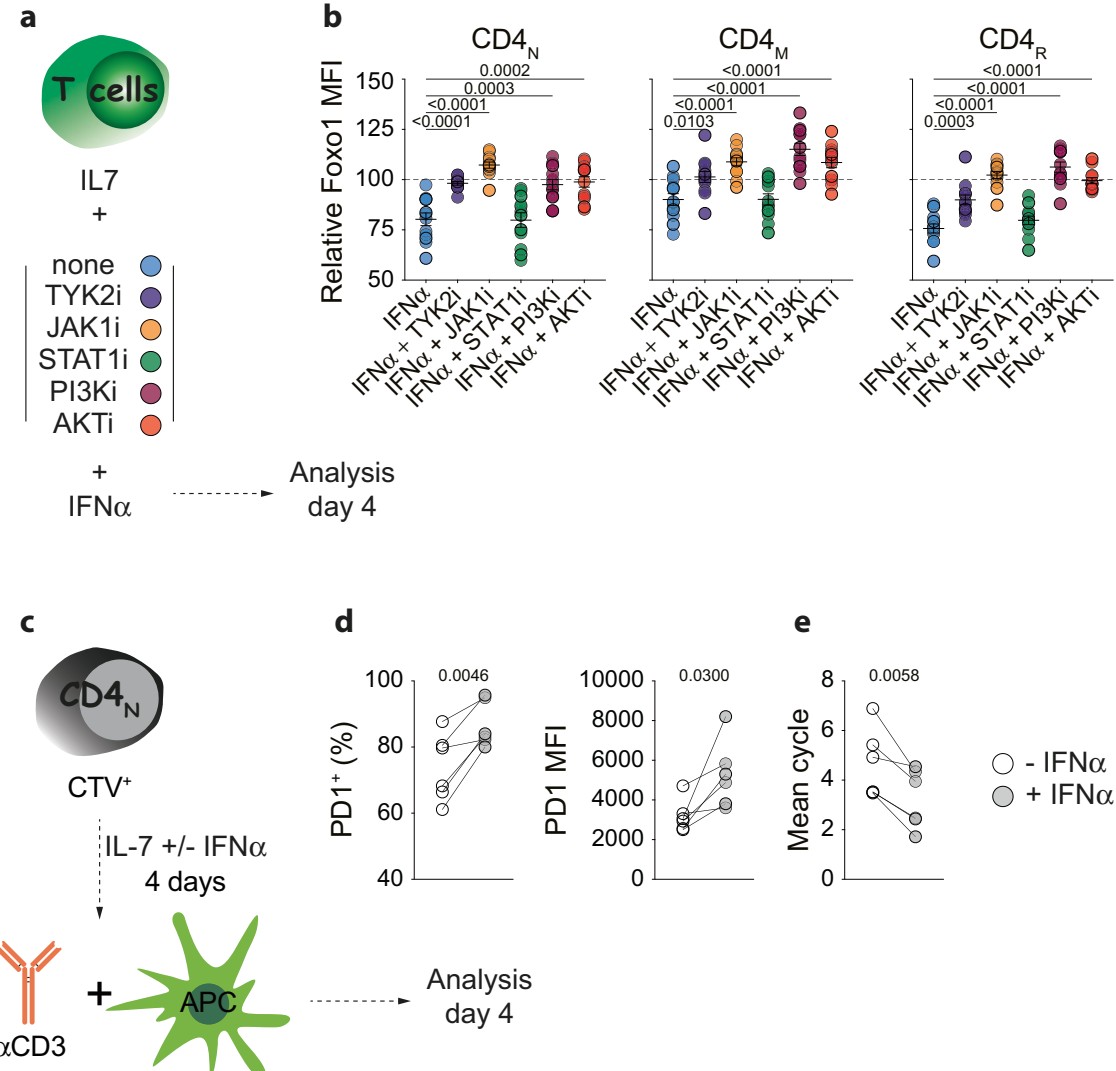

**Fig. 7 | Type 1 interferons induce Foxo1 down-regulation in T cells. a** Purified T cells from young WT mice were cultured for 4 days with IL-7 alone or together with the indicated inhibitors in the presence or absence of IFNα4. Diagram illustrating the experimental model. **b** Relative Foxo1 MFIs for CD4$_N$, CD4$_M$ and CD4$_R$ cells recovered after 4 days of culture with IL-7 in the presence of IFNα4 and the indicated inhibitors. Relative MFIs were calculated after barcoding by dividing the MFI of a given T-cell subset in the presence of one given inhibitor and IFNα4 by the MFI of the same T-cell subset cultured with the same inhibitor alone. Statistics were calculated to compare Foxo1 down-regulation induced by IFNα4 alone with Foxo1 down-regulation induced by IFNα4 in the presence of the indicated inhibitors. **c**−**e** CTV-labeled purified CD4$_N$ cells from WT mice were precultured or not with

IFN-α and then stimulated for 4 days with anti-CD3 in the presence of splenocytes from CD3ε$^{KO}$ mice. Diagram illustrating the experimental model (**c**). Percentages of PD1$^+$ cells and PD1 MFI among the progeny of CD4$_N$ cells upon in vitro activation (**d**, $n = 6$ independent experiments). The average number of cell cycles undergone by CD4$_N$ cells upon in vitro activation was calculated and plotted (**e**, $n = 6$ independent experiments). Each pair of dots represents an individual experiment. Quantifications are represented as Means ± SEM. The significance of differences between two series of results was assessed using Student's unpaired (**b**) or paired (**d**, **e**) $t$ test. Significant ($p < 0.05$) $p$-values are indicated. Source data are provided as a Source Data file.

## Fluorescence staining and flow cytometry

Cell suspensions were collected and dispensed into 96-well round-bottom microtiter plates (Greiner Bioscience; $6 \times 10^6$ cells/well). Surface staining was performed as described in ref. 98. Briefly, cells were incubated on ice, for 15 min/step, with antibodies (Abs) in 5% FCS (Biochrom), 0.1% NaN3 (Sigma-Aldrich) in PBS. Each cell staining reaction was preceded by a 15-min incubation with a purified anti-mouse CD16/32 Ab (FcgRII/III block; 2.4G2, BioXcell). The Foxp3 Staining Buffer Set (eBioscience) was used for Foxp3, Foxo1, Foxo3, Ki-67, TOX, and TCF-1 intracellular staining. This protocol was also used to determine intra + extracellular expression of CCR7. To assess pFoxo and pAKT levels ex vivo, spleen cells were immediately fixed in 4% PFA for 5 min at 37 °C. Cells were then washed and permeabilized by adding ice-cold 100% methanol to a final concentration of 90% methanol and

incubated for at least 30 min at −20 °C. After extensive washing, cells were barcoded using anti-CD45 antibodies and stained for cell surface antigens. After an additional wash, intracellular antigens, including Phospho-Foxo1 (Thr24)/Foxo3 (Thr32) or Phospho-AKT (Ser473) (both from Cell Signaling), were stained overnight at 4 °C. For determination of intracellular cytokine production, cells were stimulated with 0.5 μg/ml PMA, 0.5 μg/ml ionomycin, and 10 μg/ml brefeldin A (all from Sigma-Aldrich) for 2 h at 37 °C. Cells were then stained for surface markers, fixed in 2% paraformaldehyde in PBS, and permeabilized with 0.5% saponin, followed by labeling with specific cytokine Abs. Antibodies and dilutions are listed in Supplementary Data 7. Multi-color immunofluorescence was analyzed using a BD-Fortessa cytometer (BD Biosciences). Data acquisition and cell sorting were performed at the Cochin CYBIO facility.

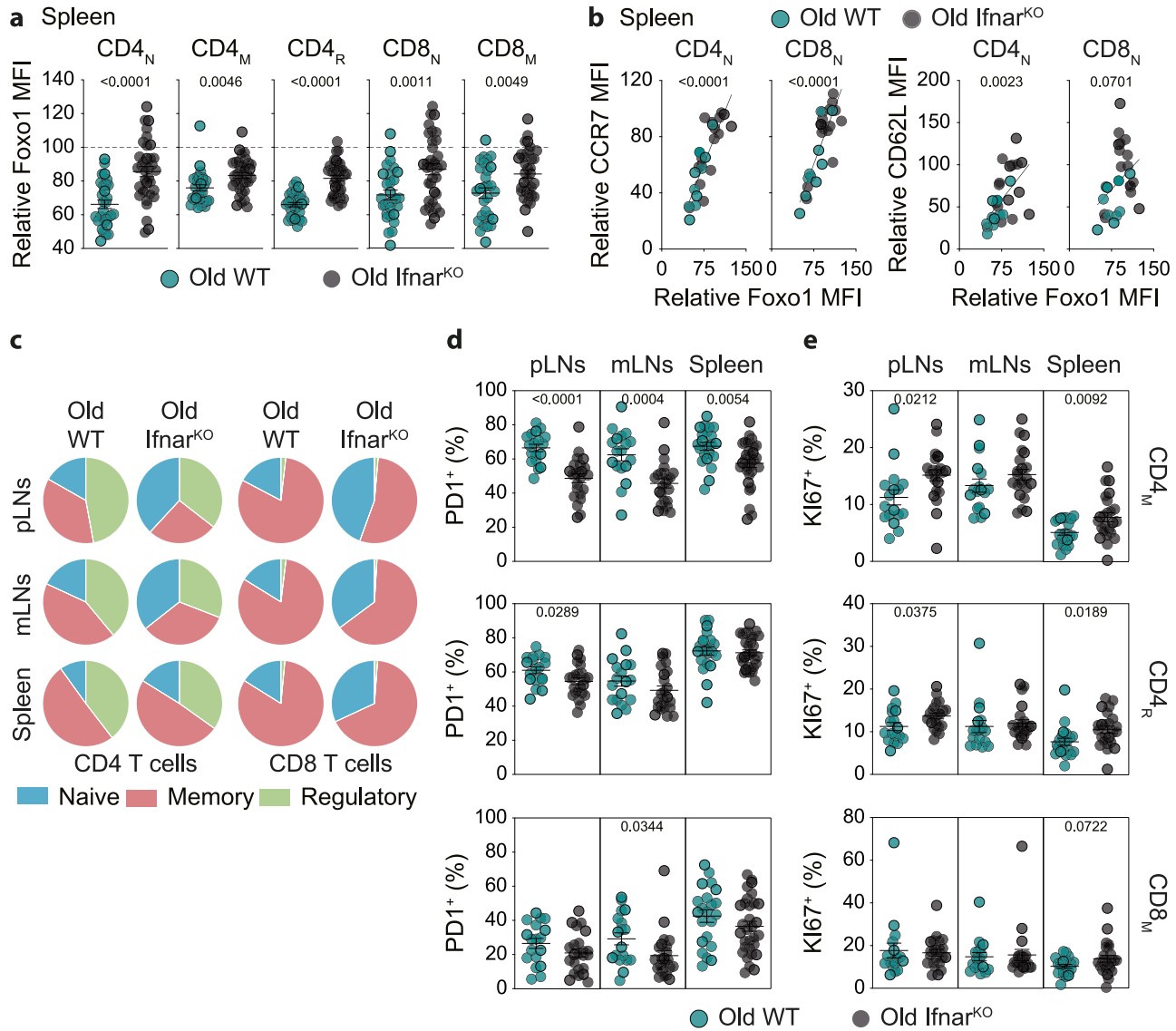

**Fig. 8 | Analysis of the T-cell compartment of old Ifnar^KO mice. a** Cell suspensions from the spleen of a 3-month-old WT mouse (Young adult), a 22-month-old WT mouse (Old) and a 22-month-old Ifnar^KO mouse (old) were first stained separately with anti-CD45 antibodies conjugated to different fluorochromes and then mixed before further staining. Relative Foxo1 MFIs were calculated by dividing the MFI of a given T-cell subset of old mice by the MFI of the same T-cell subset of the barcoded young adult WT mouse. **b** Relative expression of CCR7 and CD62L by CD4_N and CD8_N cells from the spleen of old mice as a function of their Foxo1 relative expression. Relative MFIs were calculated by dividing the MFI of a given T-cell subset of old mice by the MFI of the same T-cell subset of the barcoded young adult

WT mouse. **c** Distribution of CD4_N, CD4_M and CD4_R cells among CD4 T cells and of CD8_N, CD8_M, and CD8_R cells among CD8 T cells recovered from the indicated SLOs of old WT versus old Ifnar^KO mice. Percentages of PD1^+ (**d**) and KI67^+ (**e**) cells among CD4_M, CD4_R and CD8_M cells recovered from the indicated SLOs of 22-month-old WT versus 22-month-old Ifnar^KO mice. Quantifications are represented as Means ± SEM. The significance of differences between two series of results was assessed using Student's unpaired $t$ test (**a**, **d**, **e**). For assessing correlations, Pearson correlation (two-sided test, coefficient and 95% confidence intervals) was used (**b**). Significant ($p < 0.05$) or almost significant ($0.05 < p < 0.10$) $p$-values are indicated. Source data are provided as a Source Data file.

## Imaging flow cytometry

LNs from C57BL/6 Foxp3-GFP were harvested and fixed immediately in 4% paraformaldehyde. Then, cells were labeled with anti-CD4 and anti-CD44 Abs, permeabilized, and stained for Foxo1 and H3 as described above. GFP staining was used to identify CD4R cells. H3 was used to stain the nucleus. Cells were acquired with ImageStreamX (Amnis; EMD Millipore) equipped with four lasers (375 nm (70 mW), 488 nm (200 mW), 561 nm (200 mW), and 642 nm (150 mW)). Laser intensities were set to maximal values that do not saturate the camera. Cells were gated for single cells using the area and aspect ratio features, and for focused cells using the gradient root mean square feature. Data were analyzed with IDEAS 6.2 software. Foxo1 mean fluorescence intensities

were calculated within masks created on H3 (nucleus), CD4 (Cells), and cytoplasm (Cells−nucleus combined mask).

## Adoptive transfer of CD4 T cells

CD4 T cells were purified from pLNs + mLNs or thymi of the indicated mice by incubating cell suspensions on ice for 20 min with a mixture of anti-CD11b (Mac-1; BioLegend), anti-CD19 (1D3, BioXcell), anti-Ter119 (TER-119, BioLegend) and anti-CD8α (53-6.7) Ab obtained from hybridoma supernatant and then with magnetic beads coupled to anti-rat Igs (Dynal Biotech). Then, in some experiments (Fig. 2f–h; Fig. S4d, e; Fig. S7c–e), CD4_N cells were sorted by flow cytometry as CD44^{-/low} Foxp3-GFP^- Lineage^- (NK1.1^- TCRγδ^- CD8β^- CD11b^- CD11c^- CD19^-

CD25⁻) cells using a FACSAria III flow cytometer (BD Biosciences) and injected i.v. into recipient mice.

## Intrathymic injection of FITC

Mice were anesthetized with 3% isoflurane in air for induction and maintained with 1.5%, then depilated in the thoracic region and placed in supine position on a dedicated heating platform, allowing monitoring of ECG, temperature and respiratory frequency. Ultrasound-guided injection was performed on anesthetized mice using a Vevo 2100 high resolution ultrasound device (Visualsonics, Toronto, Canada) with a 40 MHz probe (MS-550) and its dedicated injection mount. A Hamilton syringe (1705TLL) connected to a 27G 19 mm needle was used to ensure proper control on the injected volume. The Visualsonics system was then used to guide the needle (using B mode imaging) into the part of the thymus targeted and make sure the injection was properly performed. As previously described in ref. 101, 10 µl of FITC (5 mg/ml in PBS, Sigma-Aldrich) was injected into each thymic lobe. The thymus, spleen, and LNs (pooled pLNs and mLNs) were recovered 16 h later.

## Cell cultures

For Figs. 4, S7, $5 \times 10^5$ CD4$_N$ or CD8$_N$ cells, sorted by flow cytometry were first labeled with a 5 mM CellTrace violet proliferation kit (Invitrogen), according to the manufacturer's guidelines and then cultured 2 or 4 days in the presence of soluble anti-CD3 Abs (145−2C11; 0.1 µg/mL produced in our laboratory) and $25 \times 10^4$ splenocytes from C57BL/6 CD3ε$^{KO}$ mice. For Figs. 6c, S10b, T cells were purified from LNs of C57BL/6 Foxp3-GFP by incubating cell suspensions on ice for 20 min with a mixture of anti-CD11b (Mac-1; BioLegend), anti- CD19 (1D3, BioXcell) and anti-Ter-119 (TER-119; BioLegend) Abs and then with magnetic beads coupled to anti-rat Igs (Dynal Biotech). Then, $10^5$ cells were cultured in the presence of IL-7 (10 ng/ml; R&D Systems) alone or together with one of the following cytokine: IFNα4 ($10^5$ U/ml[99];), IFNγ (25 ng/ml; Sigma-Aldrich), IL-1β (25 ng/ml; R&D Systems), IL-2 (15 ng/ml; R&D Systems), IL-4 (25 ng/ml; R&D Systems), IL-6 (25 ng/ml; Invitrogen), IL-12 (25 ng/ml; R&D Systems) and TNFα (25 ng/ml; R&D Systems). For Figs. 7b, S10d, $10^5$ purified T cells were cultured in the presence of IL-7 (10 ng/ml; R&D Systems) alone or together with one of the following inhibitor: Deucravacitinib (TYK2i; 0,625µM; Selleckchem), Upadacitinib (JAK1i; 0,0625 µM; Selleckchem), Fludarabine (STAT1i; 3,125 µM; Selleckchem), Alpelisib (PI3Ki; 5 µM; MedChemExpress), AKT inhibitor VIII (AKTi; 0,75 µM; Calbiochem) with or without IFNa4 ($10^5$ U/ml).

## Microarrays

CD4 T cells from LNs (pooled peripheral and mLNs) of the indicated mice were enriched as previously described in ref. 98. Then, CD4$_N$ cells were flow cytometry sorted using a FACS-ARIA3 flow cytometer as described above. Total RNA was extracted using the RNeasy Mini kit (QIAGEN). After validation of the RNA quality with Bioanalyzer 2100 (using Agilent RNA6000 pico chip kit), 20 ng (E18-052) and 700 pg (E20-015) of total RNA is reverse transcribed following the Ovation Pico WTA System V2 (Tecan). Briefly, the resulting double strand cDNA is used for amplification based on SPIA technology. After purification according to Tecan protocol, 3.6 µg of Sens Target DNA are fragmented and biotin labeled using Encore Biotin Module kit (Tecan). After control of fragmentation using Bioanalyzer 2100, cDNA is then hybridized to GeneChip® ClariomS Mouse (Affymetrix) at 45 °C for 17 h. After overnight hybridization, chips are washed on the fluidic station FS450 following specific protocols (Affymetrix) and scanned using the GCS3000 7G. The scanned images are then analyzed with Expression Console software (Affymetrix) to obtain raw data (cel files) and metrics for Quality Controls. RMA normalization[102] is performed using R. Entrez Gene CDF of Brain Array was used for normalization. Transcriptomic signatures of CD4$_N$ cells from old WT mice and young

adult mice deficient in Foxo1 expression have been deposited in the Gene Expression Omnibus at http://www.ncbi.nlm.nih.gov/geo/ (accession numbers GSE211827 and GSE211365 respectively).

## Next generation sequencing

CD4$_N$ cells from LNs (pooled peripheral and mLNs) of the indicated mice were flow cytometry sorted using a FACS-ARIA3 flow cytometer and then stimulated for 4 days with an anti-CD3 antibody in the presence of splenocytes from CD3ε$^{KO}$ mice. Fastq files were then aligned using STAR algorithm (version 2.7.6a), on the Ensembl Mus musculus GRCm38 reference, release 101. Reads were then count using RSEM (v1.3.1) and the statistical analyses on the read counts were performed with R (version 3.6.3) and the DESeq2 package (DESeq2_1.26.0) to determine the proportion of differentially expressed genes between two conditions. We used the standard DESeq2 normalization method (DESeq2's median of ratios with the DESeq function), with a pre-filter of reads and genes (reads uniquely mapped on the genome, or up to ten different loci with a count adjustment, and genes with at least ten reads in at least three different samples). Following the package recommendations, we used the Wald test with the contrast function and the Benjamini-Hochberg FDR control procedure to identify the differentially expressed genes. R scripts and parameters are available on Github, https://github.com/GENOM-IC-Cochin/RNA-Seq_analysis/releases/tag/v1.202112. The transcriptomic signature of activated CD4$_N$ cells from young adult mice deficient in Foxo1 expression has been deposited in the Gene Expression Omnibus (NCBI) at https://www.ncbi.nlm.nih.gov/geo/query/acc.cgi?acc=GSE210191.

## Calculations

For Figs. 4d, S7d, we estimated the CTV dilution factor (f) due to stimulation, where f = CTV mean fluorescence intensity (MFI) in the absence of stimulation/CTV MFI in the presence of stimulation. Then, because the intracellular amount of CTV is halved during each cell cycle, the average number of cell divisions was calculated as Mean cycle = log2(f).

## Statistical analysis

Data are expressed as mean ± SEM, and the significance of differences between two series of results was assessed using the Student's unpaired or paired $t$ test. For assessing correlations, Pearson correlation (two-sided test, coefficient, and 95% confidence intervals) was used. Values of $p < 0.05$ were considered as statistically significant.

## Reporting summary

Further information on research design is available in the Nature Portfolio Reporting Summary linked to this article.

# Data availability

Transcriptomic signatures of CD4$_N$ cells from old WT mice and young adult mice deficient in Foxo1 expression have been deposited in the Gene Expression Omnibus under accession numbers GSE211827 and GSE211365, respectively. The transcriptomic signature of activated CD4$_N$ cells from young adult mice deficient in Foxo1 expression has been deposited in the Gene Expression Omnibus under accession number GSE210191. Source data are provided with this paper.

# Code availability

R scripts and parameters are available on Github https://github.com/GENOM-IC-Cochin/RNA-Seq_analysis/releases/tag/v1.202112.

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

## Acknowledgements

We greatly acknowledge K. Bailly and M. Andrieu from the Cochin Immunobiology facility. We thank C. Randriamampita, N. Bercovicci, and J. Helft for critical reading of our manuscript. This work was supported by a grant from the "Fondation pour la Recherche Médicale" (FRM team number EQU202103012662). Théo Level is supported by a Ph.D. fellowship from the "LaBex Who AM I?". The authors dedicate this manuscript to the memory of Dr. Arnaud Delpoux, a former Ph.D. student of the laboratory.

## Author contributions

A.D., N.B., and B.L. designed experiments. A.D., N.B., A.L., V.G., M.G., C.M., G.R., C.C., A.L.B., B.M., and C.A. did the experiments. A.D. and N.B. analyzed the flow cytometry data. T.L., S.J., F.L., M.L.G., and B.L. analyzed the transcriptomic data. A.D. and C.A. made the figures. B.L. wrote the paper.

## Competing interests

The authors declare no competing interests.
