## [Peer Review File · Nature Communications]

REVIEWER COMMENTS

Reviewer #1 (expert in immune ageing and T cell function in old age):

Here Durand et al. explored the role of Foxo transcription factors in T cell aging using naturally aging mice and T cell-specific Foxo1 deletion models. They demonstrate that T cell expression of Foxo1 gradually declines with age and clearly showed that the age-related changes in the host microenvironment drive this phenotype. The authors further demonstrate that elevated type-I IFN signaling in old hosts underlies the Foxo1 down-modulation in T cells, and that loss of Foxo1 activity reprograms naïve T cells to acquire more differentiation memory states and exhausted phenotype. Overall, this study provides new and potentially interesting information on T cell aging by demonstrating that the Type I IFN-Foxo1 axis at least in part drives the age-associated proportional and phenotypic changes in T cells in secondary lymphoid organs.

Strengths. Overall, the strength of the study is that the authors used elegant mouse models to selectively delete Foxo1 in T cells (CD4Cre.fl-stop-fl-Foxo1; Foxo1TKO mice). There are also proper comparisons of age-, sex-, and genotype- controls, adoptive transfers and tracking of Foxo1fl/fl and Foxo1TKO cells in different SLO, and transcriptional comparisons of adult and old T cells and adult Foxo1fl/fl and Foxo1TKO T cells. In addition, the authors also did a nice job to compare and overlap their transcriptional data to publically available mice or human data sets to establish the similarities and differences in the signatures of Foxo1TKO T cells and old WT T cells.

Limitations: As presented, the study has several limitations and overinterpretations that require additional supporting data or reformulation of the text. The study is short on important experimental details that hamper interpretation of physiological impact. Second, while the IFN-I and Foxo1 connection has been reasonably substantiated, the exhaustion “profiles” need more rigor both experimentally and in interpretation. Finally, the limits of the IFN-I – Foxo1 axis require loss-of-function testing, because it is not clear whether all of the IFN-I effects are Foxo1-mediated..

Major specific comments.

1. Phenotypes used to define cell subsets need to be discussed immediately in Fig. 1 and in the text. As is, using CD44 as the sole defining marker for T_n and T_m cells is insufficient. I appreciate that some of the conventional markers like CD62L and CCR7 are reduced as a consequence of Foxo1 loss, but they are not negative. At least a few critical experiments need to be repeated (or shown, if they have been performed) with additional markers.
2. Figure 1. FMO stain controls for Foxo1 and Foxo3 are needed to show the background signal or if any spillover.
3. Fig 1A, some CD8_n and m cells exhibit increased Foxo1 with aging. What is the biological significance of this heterogeneity? Do these CD8_{Tn} Foxo1^{hi} cells survive better in vivo?
4. Both T_m and T_r cells lose Foxo1 even more profoundly than T_n cells, yet they accumulate in SLO. What is the basis for this differential survival?
5. The authors claim that naïve Foxo1TKO T cells have a higher propensity to acquire memory phenotype in the aged host, however, they did not address if the homeostatic proliferation of these cells in response to trophic cytokine stimulation drives them to acquired virtual-memory phenotype, as judged by lack of CD49d expression on T_{cm} cells. BrdU experiments would be very informative in that regard.
6. Related to the above, is the main effect of Foxo1 loss on T_n cells to downregulate CD62L and CCR7 and hamper their homing to LN? That would potentially force T_n cells to seek other means of maintenance, including the upregulation of the IL-15R, and may redirect them to bone marrow from the SLO. Also, Foxo1 deficiency in the environment appears to further pronounce the deficiency in T cells (Fig. 2g,h) – are there effects of Foxo1 deletion on SLO, given the recent literature on SLO aging (Richter, J. et al, PlosPathogens, 2015; Becklund, B. et al, Sci. Rep. 2015, Thompson, H.L. et al., Aging Cell 2019; Sonar, S., et al., PNAS USA 2022]? And who is the donor in Fig. 2h?
7. In figure 1c,d the authors showed that both the nuclear to cytoplasmic localization of Foxo1 is reduced in old T cells, and reasoned that Foxo1 downregulation is mainly driven by its transcription levels. However, Foxo1 activity is mainly regulated by its phosphorylation status. Are there any changes in the phosphorylation status of Foxo1 in aged T cells?
8. Figure 2d shows that at 1 mo, Foxo1TKO and Foxo1control mice have comparable thymic cellularity and SP CD4 and CD8 T cells. To strengthen the view that Foxo1TKO mice do not exhibit

a defect in thymic function, the authors need to analyze the frequency and numbers of DN stages and DP cells and plot the correlation of T cells in SLO and DP thymocyte numbers.

9. Intrathymic FITC-labeling in 1 mo old Foxo1TKO and Foxo1control mice showed that recent thymic emigrants had a defect in homing specifically to the LNs but not the spleen, and given the data that Foxo1 regulates the expression of lymphoid homing molecules, CCR7, CD62L, and S1P1, do authors detect increased numbers of Foxo1TKO cells in the circulation as a result of their inability to seed or home LNs?

10. The authors need to analyze the exhausted T cell phenotype throughout the manuscript by employing a co-expression of PD-1, TIGIT, and CD39 markers that they used in isolation. In fact, connections between TIGIT and CD39 and Foxo1 are weak as presented, and even inverted in some cases in Fig. 3. PD-1 is an activation, as much as an exhaustion, marker, and using it singularly to define exhaustion is inappropriate. Also, when analyzing the correlation of Ki67+ proliferating cells or FOXO1 level or function of age with T cell exhausted phenotype, the authors need to clearly describe how they define exhausted T cells.

11. Figure 4g. The authors compared the transcriptomic signature of Foxo1TKO and Foxo1control T cells by overlapping them onto in vitro T cell activation data from adult and old humans. The d5 in vitro activated T cells do not necessarily reflect a true exhaustive signature, instead, they show TCR and co-stimulation related signatures, in such as case the authors need to modify their conclusion of this data, as throughout the description of their figure 4, they highlight that activated Foxo1TKO T cells exhibit exhaustive signature, which is perfectly overlapped with the human in vitro stimulated T cell transcriptional signature.

12. Extended data Figure 4 is not cited in the text.

13. In Figure 5, the authors recover 0.1-1% of all injected cells. This provides substantial opportunity for "survival bias" and needs to be controlled for.

14. In Figure 6, cytokine concentrations need to be indicated and discussed. Are they physiological or not? ; moreover, cell recoveries after culture need to be shown as well, with IL-7 and with all other cytokines added. This again would speak to the presence or the absence of the "survival bias" in cultures.

15. Are all the effects of IFN-I mediated via Foxo1? It has long been known that IFN-I has an antiproliferative effect, so showing that there is reduced proliferation in its presence is certainly not surprising, but it operating via Foxo1 to mediate this effect would be surprising. The authors should treat Foxo1-KO or Foxo1-sufficient cells with IFN-I and document linearity of the pathway (or not) to address this issue?

16. In Figure 7A, a drop in Foxo1 MFI in old Ifnar^{-/-} mice compared to 3 mo adults suggest that in addition to Type I IFN signaling, there is a possibility of other additional contributing factor(s) in the aged microenvironment that regulates FOXO1 level, the authors need to discuss this.

Other comments:

- There are many imprecise points or missing data, the manuscript would benefit from a thorough proofing and from providing as much detail as possible

- Ln 178, "important" difference should be significant difference

- Ln 181, "reduced by 2" is left to the reader to interpret. 2 what? Fold? %?

- Fig. 1C, panels should be labeled to denote young adult vs old

- Throughout the manuscript, the 3mo old mouse should be called young adult. Young could be anything from infant to 6 months, and these are sexually mature young adult mice.

- Ln 862 - "blood cells were barcoded with cells" - this is makes no sense

- Ln 290, please reference prior work on conversion of Tn cells into T virtual memory cells with aging (Rudd, B.D., et al., PNAS 2011; Decman, V. et al., J. Immunol, 2012; Chiu, B-C. .. Chensue, S., J. Immunol, 2013; Renkema, K.R. et al., J. Immunol, 2014).

-

Reviewer #2 (expert in inflamm-ageing and T cell function in old age):

Durand et al convincingly show that FOXO1 but not FOXO3 is downregulated in T cells with progressive age and that FOXO1 deficiency in T cells from knockout mice reproduces many of the age-associated transcriptional and functional changes. In adoptive transfer experiments they could show that FOXO1 deficiency is cell extrinsic, induced by type I interferons and possibly other

cytokines. The experiments use appropriate mouse knock-out models, and the results are generally convincing. Given the importance of FOXO1 in T cell homeostasis and T cell differentiation, the significance of the findings is high, and the identification of an upstream mediator has potential impact. On selected topics, a more thorough examination or at least discussion of molecular mechanisms would be desirable.

Specific comments

1. The authors suggest that the major mechanism of low FOXO1 is transcriptional and not degradation based on their finding that they did not find an increased accumulation of FOXO1 in the cytoplasm vs the nucleus. Generally, FOXO1 concentrations in the context of aging are lower due to degradation. Increased AKT activation appears to also play a role for low FOXO1 in old T cells after activation (Jin et al, Science Advances 2020). The ratio of cytoplasmic/nuclear FOXO1 may not be the most conclusive. Also, FOXO1 degradation would reduce FOXO1 transcription. Phosphorylation states of FOXO1 and upstream kinases such as Akt may be more informative. Also, is the low FOXO1 reversible upon Akt inhibition?
2. IFN reduced FOXO1 transcription within two hours. What is the mechanism? Is this affected by JAK or PI3K/Akt inhibition?
3. FOXO1 has complex effects on the differentiation of T cells into different functional subsets such as effector T cells vs memory T cells or TFH and Tregs. The authors convincingly show that naïve cells are depleted at the expense of memory cells, which may reflect its effect on cellular quiescence rather than differentiation. How similar is the subset distribution of differentiated T cells in old and Foxo1 deficient T cells?
4. It is unclear whether FOXO1 deficiency truly induces exhaustion, and the authors should discuss the limitation of their activation system to make conclusions on exhaustion. FOXO1 deficiency favors effector over memory cell differentiation, which may be sufficient to explain the observed transcriptional signatures. TIM3 is negative in their studies, other co-inhibitory receptors could be a consequence of activation. In addition, lysosome dysfunction in FOXO1-deficient T cells could contribute to the increased cell surface expression of negative regulatory receptors (see also Jin et al., Science Immunology 2021), even if cells are not exhausted.

Reviewer #3 (expert in Foxo transcription factors):

This manuscript reports Foxo1 downregulation as one of the mechanisms of T cell aging. The authors showed Foxo1 is down-regulated with age in mouse T cells. This leads to an increased differentiation of naïve T cells into memory T cells. Those aberrantly differentiated T cells express higher exhaustion markers. They further demonstrated that age-dependent T cell aberrations are cell extrinsic through adoptive transfer experiments. Aging recipient tissues predispose young donor-derived T cells to exhaustion and alter their fate. The authors further identified type-1 interferons as a driver of Foxo1 down-regulation and T cell aging. They support the claim with further evidence: Infa deficiency is sufficient to prevent decreased Foxo1 expression, enrichment of CD4M cells at the expense of CD4N cells, acquisition of an exhausted phenotype, and decreased turn-over.

Overall, the data are consistent with the conclusions and extended data also further supports the conclusion. As the authors discussed, the previous reports indicated the age-progressive downregulation of Foxo1 in human T cells and other cell types, and thus the present report is not entirely surprising. Regardless, I find the report to have solid and systematic analysis and believe it should be a good candidate for publication once the authors address the remaining concerns. The mechanisms by which (1) IFN- α or other pro-inflammatory cytokines reduce Foxo1 expression and (2) Foxo1 deficiency increases the expression of exhaustion markers, in particular, could enhance the significance of the current study.

Specific points

Fig 1a. The method of prelabeling samples with anti-CD45 and combining them for further FOXO staining is a rigorous way of determining the protein expression. However, CD45 expression may be subject to age and/or FOXO activity. Please present evidence that this is not the compromising

factor in this assay.

Fig 2: Earlier studies repeatedly showed Foxo1 is necessary for the development of Treg population. Foxo1-TKO and Foxo1-low aging T cells are skewed toward Treg differentiation in this study. Authors should discuss the potential mechanisms explaining the differences.

Extended Fig. 2a: The effect of Foxo1 absence in CD8 T cells is impressive. Foxo1-TKO is limited to CD4 T cells. How do Foxo1-deficient CD4 T cells influence CD8 T cell fate? Similarly, PD1 expression on CD8 T cells is significantly lower in Foxo1-TKO (Fig. 3c). Are these due to cell intrinsic or extrinsic mechanisms?

Fig. 3c and Extended Data Fig. 3c: Is the regulation of exhaustion markers transcriptional? Do authors test individual marker expression by qRT-PCR? Also, what is the mechanism of Foxo1-deficiency induced exhaustion marker induction? Is it direct de-repression by Foxo1 depletion or secondary to the exhaustion program and thus just the correlation? Several earlier studies reported either maintenance of PD-1 expression by Foxo1 (PMID: 25464856) or lack of PD-1 induction in Foxo1 knockout T cells (PMID: 33503413). These various responses might be due to the heterogeneous cellular context and non-direct regulation by Foxo1. ChIP-seq-type analysis can help to delineate Foxo1-dependent regulation of those exhaustion markers.

Fig. 5: Is the environment-driven T-cell age phenotype a reversible phenotype? An important comparison missing is the transplantation of mid-age T cells into young mice. I doubt that an aged T cell can be rejuvenated. Instead, mid-age T cells may still benefit from a young environment and restore Foxo1 expression and normalized cell fate. This piece of evidence will definitely strengthen the conclusion on the extrinsic mechanism of T cell aging.

How does IFN-I(alpha) rapidly and selectively downregulate Foxo1 mRNA ? what is the mechanism? Without an extensive demonstration of the mechanism by which IFN- α downregulates Foxo1 transcription, it is difficult to appreciate whether the response is indirect and stoichiometric rather than programmed (and imprints a gene signature, as the authors suggest).

There are many GSEA presented throughout the figures. The gene sets and their expression changes (logFC) used in each figure should be included as supplemental tables.

Fig 6d, What is the rationale behind treating cytokines for four days rather than two hours, as Imgen did?

Is the Foxo1 activity gene signature present in young T cell gene expression? Is it reversed in IFN-I activated conditions? Can this be demonstrated by molecular methods such as ChIP-seq combined with RNA-seq?

Fig. 7, the results are largely descriptive, especially in relation to Foxo1 and Ifnar in aging T cells. Those observations can be reinforced by more mechanism-based studies.

Line 439 – CD4cre:Foxo1L/L definition is given for the first time. It should be provided at the first time it is mentioned (Fig. 1) to help readers clearly understand the experiment. The same is true for Ifnar Ko, which is shown in Fig. 7.

For the general readership, please define abbreviations such as LCMV.

POINT-BY-POINT RESPONSE TO THE REVIEWERS' COMMENTS

Reviewer #1 (expert in immune ageing and T cell function in old age):

Here Durand et al. explored the role of Foxo transcription factors in T cell aging using naturally aging mice and T cell-specific Foxo1 deletion models. They demonstrate that T cell expression of Foxo1 gradually declines with age and clearly showed that the age-related changes in the host microenvironment drive this phenotype. The authors further demonstrate that elevated type-I IFN signaling in old hosts underlies the Foxo1 down-modulation in T cells, and that loss of Foxo1 activity reprograms naïve T cells to acquire more differentiation memory states and exhausted phenotype. Overall, this study provides new and potentially interesting information on T cell aging by demonstrating that the Type I IFN-Foxo1 axis at least in part drives the age-associated proportional and phenotypic changes in T cells in secondary lymphoid organs.

Strengths. Overall, the strength of the study is that the authors used elegant mouse models to selectively delete Foxo1 in T cells (CD4Cre.fl-stop-fl-Foxo1; Foxo1TKO mice). There are also proper comparisons of age-, sex-, and genotype- controls, adoptive transfers and tracking of Foxo1fl/fl and Foxo1TKO cells in different SLO, and transcriptional comparisons of adult and old T cells and adult Foxo1fl/fl and Foxo1TKO T cells. In addition, the authors also did a nice job to compare and overlap their transcriptional data to publicly available mice or human data sets to establish the similarities and differences in the signatures of Foxo1TKO T cells and old WT T cells.

Limitations: As presented, the study has several limitations and overinterpretations that require additional supporting data or reformulation of the text. The study is short on important experimental details that hamper interpretation of physiological impact. Second, while the IFN-I and Foxo1 connection has been reasonably substantiated, the exhaustion “profiles” need more rigor both experimentally and in interpretation. Finally, the limits of the IFN-I – Foxo1 axis require loss-of-function testing, because it is not clear whether all of the IFN-I effects are Foxo1-mediated.

Major specific comments:

1. *Phenotypes used to define cell subsets need to be discussed immediately in Fig. 1 and in the text. As is, using CD44 as the sole defining marker for T_n and T_m cells is insufficient. I appreciate that some of the conventional markers like CD62L and CCR7 are reduced as a consequence of Foxo1 loss, but they are not negative. At least a few critical experiments need to be repeated (or shown, if they have been performed) with additional markers.*

New data have been added to our revised manuscript to characterize more deeply memory/effector T cells from old WT mice as well as from young adult Foxo1^{TKO} mice. In particular, as it has been shown that Foxo1 restrains CD4_N-cell differentiation into type 17 Helper (TH17) and follicular helper (TFH) T cells, we have now estimated the proportion of these 2 subsets among CD4_M cells (See our new Fig S3). In the periphery of both old WT mice and young adult Foxo1^{TKO} mice, the proportion of TH17 and TFH are significantly increased. These latest results reinforce the idea that the peripheral T-cell compartment of old WT mice and young adult Foxo1^{TKO} mice share not only quantitative similarities in terms of proportions of naïve, memory and regulatory T cells but also qualitative similarities.

As suggested by the reviewer, we have now also studied the expression of CD49d by memory T cells. Unfortunately, as explained below (see our response to comment 5), Foxo1 full deficiency strongly affects CD49d expression, even in naïve T cells.

2. *Figure 1. FMO stain controls for Foxo1 and Foxo3 are needed to show the background signal or if any spillover.*

In the figure below, FMO stain controls for Foxo1 (panel a, left), Foxo3 (panel a, right), pFoxo (panel b, left) and pAKT (panel b, right) are shown

Legends: Cell suspensions from the spleen of a 3-month-old mouse (Young adult) and a 22-month-old mouse (Old) were first stained separately with anti-CD45 antibodies conjugated to different fluorochromes and then mixed before further staining. **a**, Foxo1 and Foxo3 fluorescence histograms and the related FMO stain controls are shown for the indicated T-cell subsets of representative mice. **b**, pFoxo and pAKT fluorescence histograms and the related FMO stain controls are shown for the indicated T-cell subsets of representative mice.

3. Fig 1A, some CD8n and m cells exhibit increased Foxo1 with aging. What is the biological significance of this heterogeneity? Do these CD8Tn Foxo1-hi cells survive better in vivo?

We agree with the reviewer that with age, there is significant heterogeneity in aged mice, particularly with respect to Foxo1 expression by T cells. Notably, as shown in Fig. 5a, the extent of Foxo1 down-regulation in a given T-cell subset correlates with the levels observed in all other T-cell subsets. These results suggest that the decrease in Foxo1 expression would result from a response to an extrinsic stimulus. Fig. 6 and 7 suggest that inflammaging would induce Foxo1 down-regulation. The heterogeneity of Foxo1 expression with age could therefore reflect heterogeneity in the development of chronic inflammation with age.

Of note, we did not notice a correlation between the extent of Foxo1 down-regulation and the absolute numbers of T cells recovered from the SLOs of old mice (see figure below for CD8_N and CD8_M cells).

Spleen

Legends: Total cell numbers of CD8_N and CD8_M cells recovered from the spleen of 22-month-old mice are plotted as a function of their relative Foxo1 expression. Relative MFIs were calculated by dividing the MFI of a given T-cell subset of the old mouse by the MFI of the same T-cell subset of the barcoded young adult mouse

4. Both Tm and Tr cells lose Foxo1 even more profoundly than Tn cells, yet they accumulate in SLO. What is the basis for this differential survival?

CD4_N and CD8_N cell survival relies essentially on IL-7 availability and MHC-driven TCR tonic signaling (Jameson, 2005, seminars in immunology, 3:231; Suhr and Sprent, 2008, Immunity, 6:848; Boyman et al. 2012, Cell Mol Life Sci, 69:1597). As Foxo1 is required for maximal expression of the IL-7 receptor (Kerdiles et al, 2010, Nat Immunol, 10:176), Foxo1 down-regulation may strongly reduce naive T-cell half-life, which would at least partly explain the decreased absolute numbers of CD4_N and CD8_N cells in the SLOs of Foxo1^{TKO} mice when compared to Foxo1^{Ctrl} mice (Fig. 2 and Fig. S3). Our results also show that Foxo1-deficient naïve T cells (Fig. 2f-h) as well as WT naïve T cells transferred into old recipient mice (Fig. S7c-e) tend to differentiate more efficiently into "memory" T cells. This increased conversion of naïve T cells into memory T cells would also participate in the decrease of the naive T cell pool size while increasing the size of the memory T cell compartment.

Furthermore, IL-15 could compensate for decreased IL-7 signaling to improve CD4_M- and CD8_M-cell survival. Because CD4_R-cell survival is primarily dependent on IL-2 availability (Le Campion et al, 2012, J Immunol, 189:3339), it is not so surprising to find that CD4_R-cell homeostasis is not significantly impaired in old mice (Fig. 2c and Fig. 5e).

5. The authors claim that naïve Foxo1^{TKO} T cells have a higher propensity to acquire memory phenotype in the aged host, however, they did not address if the homeostatic proliferation of these cells in response to trophic cytokine stimulation drives them to acquired virtual-memory phenotype, as judged by lack of CD49d expression on T_{cm} cells. BrdU experiments would be very informative in that regard.

We appreciate the suggestion of the reviewer to assess CD49d expression by memory T cells. Unfortunately, it seems that there is a strong link between Foxo1 and CD49d expression:

- In CD4_N cells from Foxo1^{TKO} mice, CD49d is among the genes which expression is the more decreased at the mRNA level when compared to CD4_N cells from Foxo1^{Ctrl} mice (see figure below, panel a).
- Flow cytometry analysis of CD49d expression by CD4_N and CD8_N cells confirms CD49d down-regulation at the protein level in Foxo1-deficient naïve T cells (see figure below, panel b).
- CD49d expression by memory T cells from Foxo1^{TKO} mice is quite different from that observed in memory T cells from Foxo1^{Ctrl} mice, making it difficult to quantify and compare CD49d expression between these 2 strains (see figure below, panel c).

Legends: **a**, The relative expression of CD49d mRNAs was directly derived from the transcriptomic signature of CD4_N cells from Foxo1^{TKO} versus Foxo1^{Ctrl} mice (see Table S1). **b,c**, CD49d fluorescence histograms of the indicated T-cell subsets from the spleen of a representative Foxo1^{TKO} mouse and a representative Foxo1^{Ctrl} mouse are shown.

6. Related to the above, is the main effect of Foxo1 loss on T_n cells to downregulate CD62L and CCR7 and hamper their homing to LN? That would potentially force T_n cells to seek other means of maintenance, including the upregulation of the IL-15R, and may redirect them to bone marrow from the SLO. Also, Foxo1 deficiency in the environment appears to further pronounce the deficiency in T cells (Fig. 2g,h) – are there effects of Foxo1 deletion on SLO, given the recent literature on SLO aging (Rilchner, J. et al, PlosPathogens, 2015; Becklund, B. et al, Sci. Rep. 2015, Thompson, H.L. et al., Aging Cell 2019; Sonar, S., et al., PNAS USA 2022)? And who is the donor in Fig. 2h?

The expression of all 3 chains of IL-15R is significantly increased in CD4_N cells from Foxo1^{TKO} mice (see the figure below, panel a). However, this remains a limited increase for two of them (IL-15R α and IL-2R γ). This may compensate partially for the marked decrease in IL-7R α expression (panel b) and, as proposed by the reviewer, participate in the survival of Foxo1-deficient naïve T cells.

Legends: The relative mRNA expressions of the 3 chains of IL-15 receptor (IL-15R α , IL-2R β and IL-2R γ) (**a**) and of the IL-7R α (**b**) were directly derived from the transcriptomic signature of CD4_N cells from Foxo1^{TKO} versus Foxo1^{Ctrl} mice (see Table S1).

We have now studied the absolute numbers of T cells in the bone-marrow of 3-month-old $Foxo1^{TKO}$ and $Foxo1^{Ctrl}$ mice and did not notice an accumulation of naïve T cells within the bone marrow in mice deficient for $Foxo1$ expression in T cells (see Figure below).

Legends: Absolute cell numbers of naïve (a), memory (b) and regulatory (c) T cells in the bone-marrow of 3-month-old $Foxo1^{Ctrl}$ and $Foxo1^{TKO}$ mice.

To date, we have not studied the architecture of LNs and the spleen of $Foxo1^{TKO}$ mice because we needed all cells for staining.

Figure 2f describes the experimental model used in Figures 2g and 2h. More precisely, $CD4_N$ thymocytes from one-month-old $Foxo1^{TKO}$ or $Foxo1^{Ctrl}$ $CD45.1$ mice were transferred into $CD45.2$ mice of the same age and genotype. Thus, $CD4_N$ thymocytes from one-month-old $Foxo1^{TKO}$ $CD45.1$ mice were transferred into one-month-old $Foxo1^{TKO}$ $CD45.2$ mice and $CD4_N$ thymocytes from one-month-old $Foxo1^{Ctrl}$ $CD45.1$ mice were transferred into one-month-old $Foxo1^{Ctrl}$ $CD45.2$ mice.

7. In figure 1c,d the authors showed that both the nuclear to cytoplasmic localization of $Foxo1$ is reduced in old T cells, and reasoned that $Foxo1$ downregulation is mainly driven by its transcription levels. However, $Foxo1$ activity is mainly regulated by its phosphorylation status. Are there any changes in the phosphorylation status of $Foxo1$ in aged T cells?

First of all, we would like to thank the reviewer for his advice, as we believe it has helped us produce a much stronger manuscript. We have now studied, by flow-cytometry, the phosphorylation status of $Foxo$ and Akt . $pFoxo$ is either unchanged or decreased in T-cell subsets from the spleen of old mice but these results certainly reflect $Foxo1$ down-regulation with age. However, Akt phosphorylation is significantly increased in naïve T cells from the spleen of old mice compared to young ones (Fig. 1c). According to the literature, we found that the extent of $Foxo$ phosphorylation in all T-cell subsets correlate quite well with the extent of AKT phosphorylation confirming that AKT activation contribute to $Foxo1$ phosphorylation (Fig. 1d). However, the extent of Akt phosphorylation correlate with the extent of $Foxo1$ down-regulation only in naïve T cells (Fig. S1e). These results suggest that, at least in naïve T cells, the Akt signaling pathway may participate to $Foxo1$ down-regulation with age. These new data are described in the Results section of our revised manuscript (lines 107-117) and discussed in a paragraph of the discussion part of our revised manuscript (lines 457-467).

8. Figure 2d shows that at 1 mo, $Foxo1^{TKO}$ and $Foxo1^{control}$ mice have comparable thymic cellularity and SP $CD4$ and $CD8$ T cells. To strengthen the view that $Foxo1^{TKO}$ mice do not exhibit a defect in thymic function, the authors need to analyze the frequency and numbers of DN stages and DP cells and plot the correlation of T cells in SLO and DP thymocyte numbers.

We are now showing the absolute numbers of total, $CD4^+CD8^+$ double-positive, $CD4^+CD8^-TCR\beta^-$ triple-negative, $CD4_N$ and $CD8_N$ thymocytes from $Foxo1^{TKO}$ and $Foxo1^{Ctrl}$ mice (Fig. 2d). In agreement with the reviewer, the absolute numbers of $CD4_N$ and $CD8_N$ cells from the spleen and LNs correlate or tend to correlate (for naïve T cells from the LNs of $Foxo1^{TKO}$ mice) with the absolute numbers of double-positive thymocytes (see the figure below).

Legends: Total cell numbers of CD4_N and CD8_N cells recovered from the LNs and spleen of Foxo1^{TKO} (right) and Foxo1^{Ctrl} (left) mice are plotted as a function of the total cell numbers of double-positive (DP) thymocytes.

9. Intrathymic FITC-labeling in 1 mo old Foxo1^{TKO} and Foxo1^{control} mice showed that recent thymic emigrants had a defect in homing specifically to the LNs but not the spleen, and given the data that Foxo1 regulates the expression of lymphoid homing molecules, CCR7, CD62L, and S1P1, do authors detect increased numbers of Foxo1^{TKO} cells in the circulation as a result of their inability to seed or home LNs?

We have now measured the concentrations of T cells in the blood of 1-month-old Foxo1^{TKO} and Foxo1^{Ctrl} mice and found that the absolute numbers of both CD4_N and CD8_N cells per ml of blood were strongly decreased (see Figure below). By contrast, concentrations of CD4_M cells in the blood were sharply increased in of 1-month-old Foxo1^{TKO} mice. Thus, we did not observe increased numbers of naïve T cells in the circulation.

Legends: Absolute cell numbers of naïve (a), memory (b) and regulatory (c) T cells per ml of blood in 1-month-old Foxo1^{Ctrl} and Foxo1^{TKO} mice.

10. The authors need to analyze the exhausted T cell phenotype throughout the manuscript by employing a co-expression of PD-1, TIGIT, and CD39 markers that they used in isolation. In fact, connections between TIGIT and CD39 and Foxo1 are weak as presented, and even inverted in some cases in Fig. 3. PD-1 is an activation, as much as an exhaustion, marker, and using it singularly to define exhaustion is inappropriate. Also, when analyzing the correlation of Ki67+ proliferating cells or FOXO1 level or function of age with T cell exhausted phenotype, the authors need to clearly describe how they define exhausted T cells.

We agree with the reviewer that most markers expressed by exhausted T cells are induced in the course of T-cell activation. This is particularly true for PD1 although activated T cells express lower PD1 levels than exhausted T cells. Furthermore, PD1 is also expressed by Follicular Helper T cells (TFH). We therefore extended our study to another marker, namely TOX. Once again, this transcription factor is clearly more expressed by memory and regulatory T cells from the SLOs of old WT mice and young adult Foxo1^{TKO} mice compared to the same cells from young adult WT mice. We have also studied the expression of the transcription factor TCF-1 (encoded by the TCF7 gene) as it has been suggested that terminally differentiated exhausted T cells express lower levels of TCF-1 than exhausted T-cell progenitor cells and activated T cells. Interestingly, we found that CD4_M, CD4_R and CD8_M cells contain a significant proportion of TOX⁺ Tcf-1^{low} cells in both old WT and young adult Foxo1^{TKO} mice. All these new data are now included and discussed in the revised version of our manuscript: Fig. S5 and lines 244-255

11. Figure 4g. The authors compared the transcriptomic signature of Foxo1^{TKO} and Foxo1^{control} T cells by overlapping them onto in vitro T cell activation data from adult and old humans. The d5 in vitro activated T cells do not necessarily reflect a true exhaustive signature, instead, they show TCR and co-stimulation related signatures, in such as case the authors need to modify their conclusion of this data, as throughout the description of their figure 4, they highlight that activated Foxo1^{TKO} T cells exhibit exhaustive signature, which is perfectly overlapped with the human in vitro stimulated T cell transcriptional signature.

We apologize because we believe that the text describing these results was confusing. In the revised version of our manuscript, we have completely rewritten this paragraph (lines 289-296).

12. Extended data Figure 4 is not cited in the text.

Extended data Figure 4, now Fig. S6, is now cited in the text (lines 277 and 294)

13. In Figure 5, the authors recover 0.1-1% of all injected cells. This provides substantial opportunity for “survival bias” and needs to be controlled for.

First, we would like to sincerely apologize. In Fig. 5 and Extended Data Fig. 5 (now Fig. S7), the y-axis has been mislabeled. Indeed, the absolute numbers shown correspond to the numbers of cells recovered from the indicated SLOs per 10⁶ injected cells. This point has been corrected in all figures and figure legends. When the numbers recovered from the spleen and lymph-nodes are added, the recovery is about 75 000 cells per 10⁶ injected cells (7.5% of all injected cells, see figure below) for Fig. 5. More importantly, this recovery does not differ according to the age of recipient mice. Therefore, we do not believe that a major bias may explain our results.

Legends: 20 to 30x10⁶ purified total CD4 T cells from 2-month-old CD45.2⁺ mice were adoptively transferred into 2- or 21-month-old CD45.1/2⁺ mice. The fate of adoptively transferred cells was studied one month later. Absolute cell numbers of donor CD4 T cells recovered from the secondary lymphoid organs (Spleen + lymph nodes) of young versus old recipient mice per 10⁶ injected cells.

14. In Figure 6, cytokine concentrations need to be indicated and discussed. Are they physiological or not? ; moreover, cell recoveries after culture need to be shown as well, with IL-7 and with all other cytokines added. This again would speak to the presence or the absence of the “survival bias” in cultures.

Once again, we would like to apologize: cytokine concentrations are now indicated in the Methods part of our revised manuscript. Regarding cell recovery, to properly determine and compare Foxo1 levels after culture, a similar number of wells after a 4-day culture in the presence or absence of a given tested cytokine were mixed after labeling the cells with anti-CD45 antibodies conjugated to different fluorochromes (BV711 for the tested cytokine + IL7 versus PE-Cy7 for IL-7 alone), so that intracellular Foxo1 staining was performed in a single well. We have now re-analyzed our data to determine the proportion of CD45-BV711-labelled cells after staining (see figure below). The proportion of cells that were cultured with IFN γ , IL1 β , IL2 or TNF α was 50%, indicating that the added cytokine has not induced more cell death of cultured cells than when these cells were cultured with IL-7 alone. The proportion of cells that were cultured with IFN α , IL-4, IL-6 and IL-12 was significantly slightly below 50% indicating that these cytokines somehow induced more cell death of cultured cells than what can be observed with IL-7 alone. However, this bias in T-cell survival remains quite small and cannot explain the lower Foxo1 expression observed under certain conditions.

Legends: Purified T cells from young WT mice were cultured for 4 days with IL-7 alone or together with the indicated cytokines. Then, a similar number of wells after a 4-day culture in the presence or absence of the indicated cytokine were mixed after labeling the cells with anti-CD45 antibodies conjugated to different fluorochromes (BV711 for the tested cytokine + IL7 versus PE-Cy7 for IL-7 alone), so that intracellular Foxo1 staining was performed in a single well. The proportion of CD45-BV711-labelled cells was then determined after staining and plotted. *p < 0.05, **p < 0.01, ***p < 0.001. ****p < 0.0001, ns, not significant.

15. Are all the effects of IFN-I mediated via Foxo1? It has long been known that IFN-I has an antiproliferative effect, so showing that there is reduced proliferation in its presence is certainly not surprising, but it operating via Foxo1 to mediate this effect would be surprising. The authors should treat Foxo1-KO or Foxo1-sufficient cells with IFN-I and document linearity of the pathway (or not) to address this issue?

We agree that the experiment proposed by the reviewer would reinforce our data. So, we tried to do it. Unfortunately, the survival rate of Foxo1-deficient CD4_N cells is very low after 4 days of culture without activation (less than 30% of initial input) compared to Foxo1-sufficient CD4_N cells (around 85% of initial input) and this did not allow us to recover enough cells to perform the second part of this experiment. This decreased *in vitro* survival of Foxo1-deficient CD4_N cells certainly reflects their lower IL-7R expression (see our response to point 6).

16. In Figure 7A, a drop in Foxo1 MFI in old *Ifnar*^{-/-} mice compared to 3 mo adults suggest that in addition to Type I IFN signaling, there is a possibility of other additional contributing factor(s) in the aged microenvironment that regulates FOXO1 level, the authors need to discuss this.

We agree with the reviewer that we cannot rule out the possibility that other extrinsic factors, especially other inflammatory cytokines, may participate in the decrease in Foxo1 expression with age. The fact that in old *Ifnar*^{KO} mice, although increased compared with old WT mice, Foxo1 expression in T cells is still decreased compared with young adult WT mice indeed argues for the existence of additional contributing factors. This point is discussed in the revised version of our manuscript (lines 432-435 and line 573-575).

Other comments:

- There are many imprecise points or missing data, the manuscript would benefit from a thorough proofing and from providing as much detail as possible

- Ln 178, "important" difference should be significant difference: **Done**

- Ln 181, "reduced by 2" is left to the reader to interpret. 2 what? Fold? %?: **Done**

- Fig. 1C, panels should be labeled to denote young adult vs old:

Fig. 1c (now Fig. S1c) shows 2 examples of Foxo1 staining: on the left, a cell in which Foxo1 is localized both in the cytoplasm and in the nucleus, and on the right, a cell in which Foxo1 is mainly localized in the nucleus.

- Throughout the manuscript, the 3mo old mouse should be called young adult. Young could be anything from infant to 6 months, and these are sexually mature young adult mice: **Done**

- Ln 862 – "blood cells were barcoded with cells" – this is makes no sense: **Done**

- Ln 290, please reference prior work on conversion of Tn cells into T virtual memory cells with aging (Rudd, B.D., et al., PNAS 2011; Decman, V. et al., J. Immunol, 2012; Chiu, B-C. .. Chensue, S., J. Immunol, 2013; Renkema, K.R. et al., J. Immunol, 2014): **Done**

Reviewer #2 (expert in inflamm-ageing and T cell function in old age):

Durand et al convincingly show that FOXO1 but not FOXO3 is downregulated in T cells with progressive age and that FOXO1 deficiency in T cells from knockout mice reproduces many of the age-associated transcriptional and functional changes. In adoptive transfer experiments they could show that FOXO1 deficiency is cell extrinsic, induced by type I interferons and possibly other cytokines. The experiments use appropriate mouse knock-out models, and the results are generally convincing. Given the importance of FOXO1 in T cell homeostasis and T cell differentiation, the significance of the findings is high, and the identification of an upstream mediator has potential impact. On selected topics, a more thorough examination or at least discussion of molecular mechanisms would be desirable.

Specific comments:

1. *The authors suggest that the major mechanism of low FOXO1 is transcriptional and not degradation based on their finding that they did not find an increased accumulation of FOXO1 in the cytoplasm vs the nucleus. Generally, FOXO1 concentrations in the context of aging are lower due to degradation. Increased AKT activation appears to also play a role for low FOXO1 in old T cells after activation (Jin et al, Science Advances 2020). The ratio of cytoplasmic/nuclear FOXO1 may not be the most conclusive. Also, FOXO1 degradation would reduce FOXO1 transcription. Phosphorylation states of FOXO1 and upstream kinases such as Akt may be more informative. Also, is the low FOXO1 reversible upon Akt inhibition?*

We agree with the reviewer that our results showing that Foxo1 does not accumulate in the cytoplasm with age do not allow us to conclude that AKT and Foxo1 phosphorylations are not involved in Foxo1 down-regulation with age. Indeed, Foxo1 down-regulation is progressive with age, and when excluded from the nucleus, Foxo1 could be rapidly degraded, thus preventing its accumulation in the cytoplasm. We therefore studied the phosphorylation status of Foxo and AKT (see our new Fig. 1c). The amount of phosphorylated Foxo decreases in all T-cell subsets with age, but these results may only reflect the down-regulation of Foxo1 with age. Consistent with the literature, we found that the extent of Foxo phosphorylation correlated with the extent of AKT phosphorylation in all T-cell subsets, confirming that AKT activation may contribute to Foxo1 phosphorylation (Fig. 1d). However, phosphorylated AKT is significantly increased in CD4_N and CD8_N cells only (Fig. 1c). Interestingly, in old mice, the amount of phosphorylated AKT in these 2 T-cell subsets correlates significantly with the extent of Foxo1 down-regulation suggesting that increased AKT activation may play a role in Foxo1 down-regulation with age at least in these cells (Fig. S1e).

These new results are shown and described in the results part of our revised manuscript (lines 107-117). Together with the results we have obtained *in vitro* with inhibitors of the PI3K/AKT signaling pathway (see our response to your comment #2 below), the involvement of AKT activation in the down-regulation of Foxo1 with age is now discussed (lines 457-467).

2. *IFN reduced FOXO1 transcription within two hours. What is the mechanism? Is this affected by JAK or PI3K/Akt inhibition?*

First of all, we would like to thank the reviewer for his advice, as we believe it has helped us produce a much stronger manuscript. By using inhibitors of key kinases in our *in vitro* culture assay, we have now identified the signaling pathway downstream of the type 1 interferon receptor leading to Foxo1 down-regulation. The canonical pathway involving STAT proteins does not seem to be required for inducing Foxo1 down-regulation. In fact, in this setting, type 1 interferon-induced Foxo1 down-regulation involved the PI3K/AKT pathway. Of note, as Foxo1 controls its own expression, Foxo1 down-regulation at the protein level, will lead to its decreased transcription. These results have been added to our revised manuscript (see our new Fig. 7a,b and Fig. S9d and lines 407-415). Together with our new results on the *ex vivo* phosphorylation status of Foxo1 and AKT in old mice, they are also discussed in the discussion part of our revised manuscript: lines 457-467.

3. *FOXO1 has complex effects on the differentiation of T cells into different functional subsets such as effector T cells vs memory T cells or TFH and Tregs. The authors convincingly show that naïve cells are depleted at the expense of memory cells, which may reflect its effect on cellular quiescence rather than differentiation. How similar is the subset distribution of differentiated T cells in old and Foxo1 deficient T cells?*

New data have been added to our revised manuscript to characterize more deeply memory/effector T cells from old WT mice and from young adult Foxo1^{TKO} mice. In particular, as it has been shown that Foxo1 restrains CD4_N-cell differentiation into type 17 Helper (TH17) and follicular helper (TFH) T cells, we have now estimated the proportion of these 2 subsets among CD4_M cells from the SLOs of old mice (See our new Fig. S3c,d). In the periphery of both old WT mice and young adult Foxo1^{TKO} mice, the proportions of TH17 and TFH are significantly increased. These latest results reinforce the idea that the peripheral T-cell compartments of old WT mice and young adult Foxo1^{TKO}

mice share not only quantitative similarities in terms of proportions of naïve, memory and regulatory T cells but also qualitative similarities. A paragraph has been added in the Results part of our revised manuscript: lines 170-179.

4. It is unclear whether FOXO1 deficiency truly induces exhaustion, and the authors should discuss the limitation of their activation system to make conclusions on exhaustion. FOXO1 deficiency favors effector over memory cell differentiation, which may be sufficient to explain the observed transcriptional signatures. TIM3 is negative in their studies, other co-inhibitory receptors could be a consequence of activation. In addition, lysosome dysfunction in FOXO1-deficient T cells could contribute to the increased cell surface expression of negative regulatory receptors (see also Jin et al., Science Immunology 2021), even if cells are not exhausted.

We agree with the reviewer that most markers expressed by exhausted T cells are induced in the course of T-cell activation. This is especially true for PD1 although activated T cells express lower PD1 levels than exhausted T cells. Furthermore, PD1 is also expressed by Follicular Helper T cells (TFH). We therefore extended our study to another marker, namely TOX. Once again, this transcription factor is clearly more expressed by memory and regulatory T cells from the SLOs of old WT mice and young adult Foxo1^{TKO} mice compared to the same cells from young adult WT mice. We have also studied the expression of the transcription factor TCF-1 (encoded by the TCF7 gene) as it has been suggested that terminally differentiated exhausted T cells express lower levels of TCF-1 than exhausted T-cell progenitor cells and activated T cells. Interestingly, we found that CD4_M, CD4_R and CD8_M cells contains a significant proportion of TOX⁺ TCF-1^{-low} cells in both old WT and young adult Foxo1^{TKO} mice. All these new data are now included and discussed in the revised version of our manuscript: Figure S5 and lines 244-255.

Reviewer #3 (expert in Foxo transcription factors):

This manuscript reports Foxo1 downregulation as one of the mechanisms of T cell aging. The authors showed Foxo1 is down-regulated with age in mouse T cells. This leads to an increased differentiation of naïve T cells into memory T cells. Those aberrantly differentiated T cells express higher exhaustion markers. They further demonstrated that age-dependent T cell aberrations are cell extrinsic through adoptive transfer experiments. Aging recipient tissues predispose young donor-derived T cells to exhaustion and alter their fate. The authors further identified type-1 interferons as a driver of Foxo1 down-regulation and T cell aging. They support the claim with further evidence: Infar deficiency is sufficient to prevent decreased Foxo1 expression, enrichment of CD4M cells at the expense of CD4N cells, acquisition of an exhausted phenotype, and decreased turn-over.

Overall, the data are consistent with the conclusions and extended data also further supports the conclusion. As the authors discussed, the previous reports indicated the age-progressive downregulation of Foxo1 in human T cells and other cell types, and thus the present report is not entirely surprising. Regardless, I find the report to have solid and systematic analysis and believe it should be a good candidate for publication once the authors address the remaining concerns. The mechanisms by which (1) IFN- α or other pro-inflammatory cytokines reduce Foxo1 expression and (2) Foxo1 deficiency increases the expression of exhaustion markers, in particular, could enhance the significance of the current study.

Specific points:

1. Fig 1a: The method of prelabeling samples with anti-CD45 and combining them for further FOXO staining is a rigorous way of determining the protein expression. However, CD45 expression may be subject to age and/or FOXO activity. Please present evidence that this is not the compromising factor in this assay.

In our hands, CD45 is expressed by T cells regardless of the age of the mouse from which they were harvested. The figure below shows you 2 examples:

- a) splenocytes from a young adult mouse were labeled with BV711-conjugated anti-CD45 antibody while splenocytes from an old mouse were labeled with PE-Cy7-conjugated anti-CD45 antibody before mixing for subsequent staining steps.
- b) splenocytes from a young adult mouse were labeled with BV711-conjugated anti-CD45 antibody while splenocytes from an old WT mouse were labeled with PE-Cy7-conjugated anti-CD45 antibody and those from an old *lfnar*^{KO} mouse were labeled with BUV395-conjugated anti-CD45 antibody before mixing for subsequent staining steps.

It should be noted that very few T cells in these two examples appeared as cells not expressing CD45.

2. Fig 2: Earlier studies repeatedly showed Foxo1 is necessary for the development of Treg population. Foxo1-TKO and Foxo1-low aging T cells are skewed toward Treg differentiation in this study. Authors should discuss the potential mechanisms explaining the differences.

The previous studies dealing with CD4_R cells and Foxo1 have in fact shown a defect in the thymic development and the suppressive functions of regulatory T cells. Concerning regulatory T-cell homeostasis in the periphery, their results were not so clear:

- Kerdales et al. 2010. Immunity. 33:890 / Fig. 5D: The absolute numbers of CD4_R cells recovered from the SLOs of 8-week-old Foxo1^{TKO} mice are either equal (Spleen) or significantly higher (LNs) than those recovered from the SLOs of 8-week-old Foxo1^{Ctrl} mice.
- Ouyang et al. 2010. Nat. Immunol. 11:618 / Fig. 3b: The percentage (among CD4 T cells) and absolute number of CD4_R cells recovered from the spleen of 3-week-old Foxo1^{TKO} mice are not significantly different from those recovered from the Spleen of 3-week-old Foxo1^{Ctrl} mice. Both parameters were significantly decreased only in mice deficient for Foxo1 and Foxo3 expression compared with control mice.

In our hands, we observed a decrease in the proportion and absolute number of CD4_R in the spleen only in 1-week-old Foxo1^{TKO} mice (see Figure below):

Spleen

3. Extended Fig. 2a: The effect of Foxo1 absence in CD8 T cells is impressive. Foxo1-TKO is limited to CD4 T cells. How do Foxo1-deficient CD4 T cells influence CD8 T cell fate? Similarly, PD1 expression on CD8 T cells is significantly lower in Foxo1-TKO (Fig. 3c). Are these due to cell intrinsic or extrinsic mechanisms?

In CD4^{cre} Foxo1^{fl/fl} mice (alias Foxo1^{TKO} mice), Cre recombinase-mediated Foxo1 DNA recombination occurs at the double positive (CD4⁺CD8⁺) stage of thymic differentiation and thus affects both CD4 and CD8 T-cell lineages. This point has now been clarified in the first part of the results part of our revised manuscript (lines 127-131).

4. Fig. 3c and Extended Data Fig. 3c: Is the regulation of exhaustion markers transcriptional? Do authors test individual marker expression by qRT-PCR? Also, what is the mechanism of Foxo1-deficiency induced exhaustion marker induction? Is it direct de-repression by Foxo1 depletion or secondary to the exhaustion program and thus just the correlation? Several earlier studies reported either maintenance of PD-1 expression by Foxo1 (PMID: 25464856) or lack of PD-1 induction in Foxo1 knockout T cells (PMID: 33503413). These various responses might be due to the heterogeneous cellular context and non-direct regulation by Foxo1. ChIP-seq-type analysis can help to delineate Foxo1-dependent regulation of those exhaustion markers.

To date, we have not studied the expression of exhaustion markers by CD4_M, CD4_R and CD8_M cells from Foxo1^{TKO} mice at the transcriptional level. However, we have analyzed the transcriptional signature of CD4_N cells proficient or deficient for Foxo1 expression (Figure 1). Interestingly, mRNAs coding PD1, TIGIT and TOX (but not CD39) were already increased in CD4_N cells lacking Foxo1 expression (see Figure below).

Legends: The relative mRNA expressions of PDCD1, TIGIT, TOX and CD39 were directly derived from the transcriptomic signature of CD4_N cells from Foxo1^{TKO} versus Foxo1^{Ctrl} mice (see Table S1).

5. Fig. 5: Is the environment-driven T-cell age phenotype a reversible phenotype? An important comparison missing is the transplantation of mid-age T cells into young mice. I doubt that an aged T cell can be rejuvenated. Instead, mid-age T cells may still benefit from a young environment and restore Foxo1 expression and normalized cell fate. This piece of evidence will definitely strengthen the conclusion on the extrinsic mechanism of T cell aging.

We have now transferred CD4 T cells from 21-month-old mice into 2-month-old recipients and studied their phenotype 1 month later. Due to logistical constraints, we were unable to transfer cells from old mice into old mice expressing a different CD45 allele. However, our results strongly suggest that, with respect to the parameters

studied, T-cell aging is not reversible (Fig. S8). These results are now described in the results part of our revised manuscript: lines 351-360.

6. *How does IFN-I(alpha) rapidly and selectively downregulate Foxo1 mRNA ? what is the mechanism? Without an extensive demonstration of the mechanism by which IFN-a downregulates Foxo1 transcription, it is difficult to appreciate whether the response is indirect and stoichiometric rather than programmed (and imprints a gene signature, as the authors suggest).*

By using inhibitors of key kinases in our *in vitro* culture assay, we have now identified the signaling pathway downstream of the type 1 interferon receptor leading to Foxo1 down-regulation. The canonical pathway involving STAT proteins does not seem to be required for inducing Foxo1 down-regulation. In fact, in this setting, type 1 interferon-induced Foxo1 down-regulation involved the PI3K/Akt pathway. Of note, as Foxo1 controls its own expression, Foxo1 down-regulation at the protein level, will lead to its decreased transcription. These results have been added to our revised manuscript (Fig. 7a,b, Fig. S9d and lines 407-415).

7. *There are many GSEA presented throughout the figures. The gene sets and their expression changes (logFC) used in each figure should be included as supplemental tables.*

Gene sets and expression changes related to the transcriptomic signatures used in our manuscript have been now included as supplemental Tables:

Table S1: Transcriptomic signature of CD4_N cells deficient for Foxo1 expression (GSE211365)

Table S2: Transcriptomic signature of LCMV specific CD4 T cells 30 days after chronic versus acute LCMV infection (GSE30431)

Table S3: Transcriptomic signature of LCMV specific CD4 T cells 8 days after chronic versus acute LCMV infection (GSE30431)

Table S4: Transcriptomic signature of activated human CD4_N cells as a function of age (SRP158502)

Table S5: Transcriptomic signature of CD4 T cells 2 hours after IFN α injection (GSE75202)

Table S6: Foxo1 binding sites in CD4_N cells in gene promoter regions (\pm 1 kb from the TSS)

8. *Fig 6d, What is the rationale behind treating cytokines for four days rather than two hours, as Imgen did?*

We have studied Foxo1 expression at the protein level by flow cytometry. As we did not know anything about the half-life of the protein Foxo1, we have chosen to treat the cells for several days. We discovered later on the results of the Imgen consortium and decided to include them in our manuscript. Of note, they have studied Foxo1 expression at the mRNA level. However, we agree with the reviewer that with a decrease at the mRNA level as rapid as after 2 hours in the presence of IFN α , one can expect down-regulation of Foxo1 at the protein level to be relatively rapid. Nevertheless, in our *in vitro* culture assay, 24 hours in the presence of type 1 interferon were not enough to induce Foxo1 down-regulation at the protein level (see figure below). Foxo1 expression was even significantly increased in all T-cell subsets at that time-point.

9. *Is the Foxo1 activity gene signature present in young T cell gene expression? Is it reversed in IFN-I activated conditions? Can this be demonstrated by molecular methods such as ChIP-seq combined with RNA-seq?*

We are now showing that the transcriptional signature of CD4_N cells from Foxo1^{TKO} mice correlate quite well with the signature of CD4 T cells *in vivo* exposed to type 1 interferon (Experiments from the IMMGEN Consortium, Fig. 6h). We have also now found public data analyzing Foxo1 DNA binding sites in CD4_N cells (GSE46525; ChIPseq).

This work allowed us to compile a list of genes for which Foxo1 binds in the promoter region (\pm 1kb from the TSS; see Table S6). Interestingly, the transcription level of nearly all of these genes decreased in spleen CD4 T cells 2 hours after type 1 interferon injection (Fig. 6i). These new analyses reinforce the idea that type 1 interferons imprint CD4 T -cell transcriptional signature, in particular by decreasing Foxo1 expression. These results are now described in the results part of our revised manuscript: lines 398-405.

10. *Fig. 7, the results are largely descriptive, especially in relation to Foxo1 and Ifnar in aging T cells. Those observations can be reinforced by more mechanism-based studies.*

Figure 7 (now Fig. 8 in the revised version of our manuscript) illustrates the *in vivo* consequences of chronic exposure to type 1 interferon on T-cell aging. However, we have now added experiments aimed at deciphering the molecular mechanisms linking type 1 interferons and Foxo1 expression (Fig. 7a,b, and Fig. S9d, see our response to point 6).

11. *Line 439 – CD4cre:Foxo1L/L definition is given for the first time. It should be provided at the first time it is mentioned (Fig. 1) to help readers clearly understand the experiment. The same is true for Ifnar Ko, which is shown in Fig. 7.*

That has been done. CD4^{cre} Foxo1^{fl/fl} mice are now defined when first mentioned (Fig. 1g, line 127). The definition of Ifnar^{KO} mice has also been clarified (line 426).

12. *For the general readership, please define abbreviations such as LCMV.*

That has been done (line 514)

REVIEWER COMMENTS

Reviewer #1 (expert in immune ageing and T cell function in old age):

Durand et al. have revised and improved their manuscript on the role of Foxo transcription factors in T cell aging. Overall, the manuscript remains replete with interesting observations, many of which are relevant for the field. However, at the present, the biggest issue with this potentially important work is that the connections outlined by the authors do not appear to apply uniformly and as sweepingly as the authors contend, to all areas of T cell biology with aging. Specifically, some observations hold for CD4 but not CD8 T cells, some but not all seem to be clearly linked to Foxo1, others link IFN-I to downregulation of Foxo1 and changes in the biology of some but not other T cell subsets, and in some cases IFN-I seems to be having Foxo1-independent effects. In light of all this, the title of the manuscript needs to be re-written and conclusions re-stated, taking care to carefully specify what mechanisms they have clear evidence for, and where the evidence suggests additional complexity or lack of involvement of Foxo1 and other molecules in question. The title should be changed accordingly to a more descriptive "The role of IFN-I and Foxo1 downregulation in age-related T cell dysfunction".

MAJOR CRITICISMS:

A. Generalization between FOXO1 and aging T cell biology. Many of the observations that the authors suggest are the general consequence of the Foxo1 reduction in T cells with aging do not apply to all T cell subsets analyzed. For example, reduced steady-state proliferation, as measured by Ki-67 staining, is not Foxo1-linked in CD4R and CD8M cells (Figs. 3d and 5g), but is showing a correlation with aging. Similarly, in Figure 8 d, e, IFN-I is not linked to CD8M and CD4R expression of PD-1 and Ki-67 or to CD4M for Ki-67. Likewise, pAKT correlates to pFoxo1 in some but not other subsets (Fig. 1c). Finally, Figures 4-7 have been mostly if not exclusively performed with CD4 T cells, so whether the results extend to CD8 T cell subsets is not clear. There are no explanations or discussion of such discrepancies or heterogeneities. At the present, this is a major flaw of the manuscript as it stands. Therefore, while it appears based on the presented data that IFN-I, Foxo1 and age-related changes in some T cell subsets are connected, where and how this applies and why is far less clear. And while the TKO model is powerful to an extent, the parallels between a knockout and a physiological downregulation of a molecule are always limited.

B. Response to original critique: The authors have answered about two-thirds of my criticisms. Overall, the study remains short on important experimental details that hamper interpretation of physiological impact; figure legends and the text have to be carefully edited to describe these in sufficient details for the reader to be able to interpret the design and figures. Examples of missing data include, but are not limited to: what markers are used to define the main phenotypes - T_n, T_m and T_r; what LN were used as pLN; and what do the colors of the correlogram signify in Fig. 5. Of the specific major comments from the original critique, the points 2 (FMO, which should be shown in supplemental data), 4, 7, 8, 10, 11,12, 13, 15 and 16 are resolved. The other points (enumerated here per my original critique) still need to be answered: 1. I asked that the conventional T_n markers including CD62L and CCR7 be shown, appreciating that these may be reduced as a consequence of Foxo1 loss, but are not negative. I have also separately and elsewhere requested that CD49d (criticism #5) be included in the analysis to mark the T_{vm} cells (which are CD49d^{low}). The authors ignored the first request and stated that the second one could not be addressed because Foxo1 tightly correlated with CD49d expression – but my request was for the authors to show this analysis in wt old mice, and not Foxo1-TKO. Moreover, even for TKO cells, the figure provided in response to my question #5 shows expression (albeit lower) of CD49d in the spleen. This analysis is very important in understanding mechanisms of Foxo1-mediated changes in T cell migration with aging, and should be shown for all three molecules relative to Foxo1, and relative to one another and CD44, with CD49d differentiating between true and virtual memory; 3. I accept the author's response, and I caution them regarding the use of the term "inflammaging", which has never been quantitatively or precisely defined by their original authors; 6, and also 9. I have asked the authors to test whether the main effect of Foxo1 loss on T_n cells to downregulate CD62L and CCR7 and hamper their homing to LN. This question has not been addressed and is critical. 10. The enormous frequencies (50-100%) of T cells (particularly CD4_m and r) expressing PD-1 and also other exhaustion markers in old mice are reminiscent of true

chronic infections and are not typical of either the literature (e.g. Decman, V. et al, J. Immunol., 2012; Renkema, K. et al., J. Immunol, 2014) or of our experience in SPF mice. The authors should provide the list of pathogens excluded from (and tested for) in their colony and also the levels of CXCL-10/IP-10 in their young and old mice, as well as in representative control mice purchased and analyzed immediately upon arrival. This is also important in interpreting the driving role of IFN-I.

Other points:

- Ln. 27, "are critical at the crossroad" should be replaced by "are critically involved at..."
- Ln. 44, "the elderly" is considered a somewhat derogatory term by some individuals, and the recommended term is "older adults"
- Ln. 53, only last names of authors should be in the citations (Kenyon, 2005)
- Ln. 276, ref. Crawford is missing.
- IN light of inconsistent correlation between pAKT and pFoxo1 in Fig. 1, discussion on Ln. 461-67 needs to be softened and modified.

Reviewer #2 (expert in inflamm-ageing and T cell function in old age):

The authors are applauded for their careful revisions. I do not have further comments.

Reviewer #3 (expert in Foxo transcription factors):

The authors partially addressed previously raised points. The study remains descriptive without solid mechanisms. For example, in Reviewer Points #6 and #10, - authors concluded that type 1 interferon-induced Foxo1 down-regulation involved the PI3K/Akt pathway. However, the authors' results show AKT inhibition further suppressed FOXO1 protein expression (Fig S9d). This contradicts conventional PI3K/AKT-mediated FOXO1 regulation. What is the molecular mechanism that regulates AKT-dependent FOXO1 expression in T cells?

In addition, how viable are these CD8 T cells following treatment with inhibitors? Downregulation of FOXO1 following AKTi indicates cells have lost viability. Measuring FOXO1 expression in sick cells will surely show downregulation, regardless of actual regulation.

Aside from this confusing result, the authors do not delineate the molecular mechanism or signaling pathway that results in FOXO1 expression in aging T cells.

This manuscript requires a careful editing. For example, in lines 466-467 the sentence does not make good sense.

POINT-BY-POINT RESPONSE TO THE REVIEWERS' COMMENTS

We highly appreciate the helpful and insightful comments from all three Reviewers on our manuscript. We have now addressed all Reviewer concerns by providing new experimental data, additional information, and a revised text. The Reviewers' constructive comments have significantly contributed to the improvement of the rigor, clarity, and significance of our work. In the detailed point-by-point response below, we have addressed each concern, including additional data required to support our claims.

Reviewer #1 (expert in immune ageing and T cell function in old age):

Durand et al. have revised and improved their manuscript on the role of Foxo transcription factors in T cell aging. Overall, the manuscript remains replete with interesting observations, many of which are relevant for the field. However, at the present, the biggest issue with this potentially important work is that the connections outlined by the authors do not appear to apply uniformly and as sweepingly as the authors contend, to all areas of T cell biology with aging. Specifically, some observations hold for CD4 but not CD8 T cells, some but not all seem to be clearly linked to Foxo1, others link IFN-I to downregulation of Foxo1 and changes in the biology of some but not other T cell subsets, and in some cases IFN-I seems to be having Foxo1-independent effects. In light of all this, the title of the manuscript needs to be re-written and conclusions re-stated, taking care to carefully specify what mechanisms they have clear evidence for, and where the evidence suggests additional complexity or lack of involvement of Foxo1 and other molecules in question. The title should be changed accordingly to a more descriptive "The role of IFN-I and Foxo1 downregulation in age-related T cell dysfunction".

We thank the reviewer for these valuable suggestions. We have now revised our manuscript accordingly (see our responses below). As suggested by the reviewer, the title of our manuscript has been changed to a more descriptive title: it is now entitled "Role of type 1 interferons and Foxo1 down-regulation in age-related T-cell exhaustion".

MAJOR CRITICISMS:

A. Generalization between FOXO1 and aging T cell biology. *Many of the observations that the authors suggest are the general consequence of the Foxo1 reduction in T cells with aging do not apply to all T cell subsets analyzed.*

Throughout our revised manuscript, we have now clarified whether our observations applied to all T-cell subsets or whether differences between, for example, CD4 and CD8 T cells were observed:

* Results:

Lines 119-122: *"Consistent with the literature³², we found that the extent of Foxo phosphorylation correlated significantly with the extent of AKT phosphorylation in CD4_N, CD4_R, CD4_M and CD8_M cells and tended to correlate with the extent of AKT phosphorylation in CD8_N cells (Fig. 1d). AKT activation may thus contribute to Foxo1 phosphorylation in T cells from old mice."*

Lines 254-259: *"PD1 and TIGIT were clearly overexpressed by peripheral CD4_M and CD4_R cells from old WT or young adult Foxo1^{TKO} mice (Fig. 3c, and Fig. S5c). CD8_M cells from old WT or young adult Foxo1^{TKO} mice also exhibited higher PD1 levels than CD8_M cells from young adult WT mice although such an increase was more pronounced in old mice than in young adult Foxo1^{TKO} mice (Fig. 3c). TIGIT overexpression was observed only in CD8_M cells from old mice (Fig. S5c)."*

Lines 273-278: *"Altogether, these results strongly suggest that Foxo1 down-regulation with age in peripheral memory and regulatory T cells participate to their exhaustion. However, the overexpression of exhaustion markers is, in some cases, less marked in T cells from young adult Foxo1^{TKO} mice than in those from old WT, especially for CD8_M cells. Thus, mechanisms other than Foxo1 down-regulation may also be involved in age-related T-cell exhaustion."*

Lines 283-288: *"In all SLOs of old mice, fewer CD4_M, CD8_M and CD4_R cells expressed Ki67 (Fig. 3d). Such a decrease was not observed in CD4_R and CD8_M cells from young adult Foxo1^{TKO} mice. Indeed, the decrease was significant only in CD4_M cells in the spleen and mesenteric lymph-nodes of Foxo1^{TKO} mice. Other mechanisms not involving the down-regulation of Foxo1 expression could therefore be involved in the age-related decrease in CD4_R and CD8_M cell proliferation."*

Lines 366-369: *"Proliferation (as assessed by Ki67 expression) of donor-derived CD4_M cells was strongly decreased when transferred into aged recipient mice, compared to their proliferation when transferred into young adult recipient mice, while such decrease was not observed for CD4_R cells (Fig. 5g)."*

Lines 437-440: *"As expected, inhibitors of TYK2 and JAK1, 2 kinases directly associated with the 2 chains of the type 1 interferon receptor completely abolished (CD4 T-cell subsets) or reduced (CD8 T-cell subsets) type 1 interferon-induced down-regulation of Foxo1 (Fig. 7b and Fig. S10d)."*

Lines 445-448: “Indeed, in the presence of inhibitors of PI3K and AKT, *Foxo1* down-regulation in response to IFN- α was either completely abolished (in all T-cell subsets for the inhibitor of PI3K and in CD4 T cells for the inhibitor of AKT) or at least significantly reduced (in CD8 T cells for the inhibitor of AKT).”

Lines 475-486: “PD1 expression by CD4_M cells was significantly decreased in the SLOs of old *lfnar*^{KO} mice compared with WT mice whereas such a decrease was observed only in mesenteric lymph-nodes for CD8_M cells and in peripheral lymph-nodes for CD4_R cells (Fig. 8d). These results suggest that type 1 interferons may be key actors in inducing age-related PD1 expression in CD4_M cells but may be less involved in age-related PD1 expression in CD4_R and CD8_M cells. However, when 18-month-old WT and *lfnar*^{KO} mice were compared, PD1 expression in CD4_M, CD4_R and CD8_M cells was significantly lower in all SLOs of *lfnar*^{KO} mice compared to PD1 expression by their T-cell counterparts from WT mice (Fig. S11c). These latter results suggest that type 1 interferons may participate in the overexpression of PD1 by all T-cell subsets with age, but that other factors may compensate for the lack of expression of the type 1 interferon receptor in CD4_R and CD8_M cells in very old animals.”

Lines 489-492: “Overall, with regard to some but not all of the parameters studied, the T-cell compartment of mice lacking expression of the type 1 interferon receptor appears to be less affected by aging than the T-cell compartment of WT mice.”

**** Discussion:**

Lines 504-511: “Thus, taken together, our data suggest that the progressive decrease in *Foxo1* expression with age may first result from chronic activation of the PI3K/AKT signaling pathway induced by inflammatory cytokines at least in CD4_N and CD8_N cells. This question remains open for the other T-cell subsets as we did not observe a clear correlation between AKT phosphorylation and *Foxo1* levels in their cases. Secondly, as the transcription factor *Foxo1* increases its own transcription^{60,61}, a decrease in *Foxo1* expression at the protein level should ultimately lead to a decrease in *Foxo1* transcription.”

Lines 617-622: “However, only CD4_M cells from 22-month-old *lfnar*^{KO} mice showed a significant decrease in the expression of the key inhibitory receptor PD1. Such a decrease is observed in CD4_R and CD8_M cells only in younger animals (18-month-old mice). Thus, our results do not exclude the possibility that additional factors, in particular other inflammatory cytokines, may also contribute to T-cell exhaustion with age. It remains to identify these factors and determine whether or not they act by inducing *Foxo1* down-regulation.”

For example, reduced steady-state proliferation, as measured by Ki-67 staining, is not Foxo1-linked in CD4R and CD8M cells (Figs. 3d and 5g), but is showing a correlation with aging.

We agree with the reviewer and have amended the text to correctly reflect the results:

Lines 283-288: “In all SLOs of old mice, fewer CD4_M, CD8_M and CD4_R cells expressed Ki67 (Fig. 3d). Such a decrease was not observed in CD4_R and CD8_M cells from young adult *Foxo1*^{TKO} mice. Indeed, the decrease was significant only in CD4_M cells in the spleen and mesenteric lymph-nodes of *Foxo1*^{TKO} mice. Other mechanisms not involving the down-regulation of *Foxo1* expression could therefore be involved in the age-related decrease in CD4_R and CD8_M cell proliferation.”

Lines 366-369: “Proliferation (as assessed by Ki67 expression) of donor-derived CD4_M cells was strongly decreased when transferred into aged recipient mice, compared to their proliferation when transferred into young adult recipient mice, while such decrease was not observed for CD4_R cells (Fig. 5g).”

Similarly, in Figure 8 d, e, IFN-I is not linked to CD8M and CD4R expression of PD-1 and Ki-67 or to CD4M for Ki-67.

We agree with the Reviewer that the decrease of PD1 expression is convincing only in CD4_M cells in the SLOs of 22-month-old *lfnar*^{KO} mice compared to 22-month-old WT mice (Figure 8d). However, we are now showing data from 18-month-old mice (Figure S11c). At that age, PD1 expression in CD4_M, CD4_R and CD8_M cells was significantly lower in all SLOs of *lfnar*^{KO} mice compared to PD1 expression by their T-cell counterparts from WT mice (Fig. S11c). Type 1 interferons may thus participate in the overexpression of PD1 by all T-cell subsets with age, but other factors may compensate for the lack of expression of the type 1 interferon receptor in CD4_R and CD8_M cells in very old animals. This point has been clarified in the text of our revised manuscript:

Lines 475-486: “PD1 expression by CD4_M cells was significantly decreased in the SLOs of old *lfnar*^{KO} mice compared with WT mice whereas such a decrease was observed only in mesenteric lymph-nodes for CD8_M cells and in peripheral lymph-nodes for CD4_R cells (Fig. 8d). These results suggest that type 1 interferons may be key actors in inducing age-related PD1 expression in CD4_M cells but may be less involved in age-related PD1 expression in CD4_R and CD8_M cells. However, when 18-month-old WT and *lfnar*^{KO} mice were compared, PD1 expression in CD4_M, CD4_R and CD8_M cells was significantly lower in all SLOs of *lfnar*^{KO} mice compared to PD1 expression by their T-cell counterparts from WT mice (Fig. S11c). These latter results suggest that type 1 interferons may participate in the overexpression of PD1 by all T-cell subsets with age, but that other factors may compensate for the lack of expression of the type 1 interferon receptor in CD4_R and CD8_M cells in very old animals.”

The main conclusion of this last part of our manuscript has been also modified accordingly:

Lines 489-492: “Overall, with regard to some but not all of the parameters studied, the T-cell compartment of mice lacking expression of the type 1 interferon receptor appears to be less affected by aging than the T-cell compartment of WT mice.”

Likewise, pAKT correlates to pFoxo1 in some but not other subsets (Fig. 1c).

We thank the Reviewer for this comment. We have now updated the main text as follows:

Lines 119-122: “Consistent with the literature ³², we found that the extent of Foxo phosphorylation correlated significantly with the extent of AKT phosphorylation in CD4_N, CD4_R, CD4_M and CD8_M cells and tended to correlate with the extent of AKT phosphorylation in CD8_N cells (Fig. 1d). AKT activation may thus contribute to Foxo1 phosphorylation in T cells from old mice.”

Finally, Figures 4-7 have been mostly if not exclusively performed with CD4 T cells, so whether the results extend to CD8 T cell subsets is not clear. There are no explanations or discussion of such discrepancies or heterogeneities.

We agree with the Reviewer that our transcriptomic analyses were performed only on naïve CD4 T cells and that our adoptive transfer experiments were performed with CD4 T cells only (Figures 4 and 5). Concerning the effects of inflammatory cytokines on Foxo1 expression *in vitro*, although in Figure 6, results are presented for CD4 T cells, the results for CD8 T cells are shown in Figure S10. Similarly, analysis of the signaling pathways involved in type 1 interferon-induced Foxo1 down-regulation was carried out for both CD4 (Figure 7) and CD8 (Figure S10) T cells. In the revised version of our manuscript, we have now also compared the proliferation and phenotype of naïve CD8 T cells from Foxo1^{TKO} and Foxo1^{Ctrl} mice upon activation *in vitro*. These new results, very similar to those obtained with naïve CD4 T cells, are shown in Figure S7. However, we agree with the reviewer that some differences can be noted between CD4 and CD8 T cells and, as indicated above, throughout our revised manuscript, we have now clarified whether our observations applied to all T-cell subsets or whether differences between, for example, CD4 and CD8 T cells were observed.

At the present, this is a major flaw of the manuscript as it stands. Therefore, while it appears based on the presented data that IFN- γ , Foxo1 and age-related changes in some T cell subsets are connected, where and how this applies and why is far less clear. And while the TKO model is powerful to an extent, the parallels between a knockout and a physiological downregulation of a molecule are always limited.

We greatly appreciate the reviewer's insightful comments. We have now considerably modified our manuscript to take account of the reviewer's helpful comments.

B. Response to original critique: *The authors have answered about two-thirds of my criticisms. Overall, the study remains short on important experimental details that hamper interpretation of physiological impact; figure legends and the text have to be carefully edited to describe these in sufficient details for the reader to be able to interpret the design and figures.*

First, we would like to thank the Reviewer for this push as we think it has enabled us to produce a better manuscript. We have now addressed all the reviewers' concerns by providing new experimental data, additional information and a revised text.

Examples of missing data include, but are not limited to: what markers are used to define the main phenotypes - T_N, T_M and T_R?

We sincerely apologize for this omission. The markers used to define TN, TM and TR cells are now clearly indicated in the first paragraph of the “Results” section of our revised manuscript:

Lines 86-90: “Throughout this study, as in our previous articles ^{29,30}, regulatory CD4 T cells (CD4_R) were defined as Foxp3⁺ CD4⁺ CD8 α ⁻ TCR β ⁺ cells, memory CD4 T cells (CD4_M) as CD44^{hi} Foxp3⁻ CD4⁺ CD8 α ⁻ TCR β ⁺ cells and naïve CD4 T cells (CD4_N) as CD44^{-low} Foxp3⁻ CD8 α ⁻ TCR β ⁺ cells. CD44 expression was also used to discriminate between naïve and memory CD8 T cells (CD8_N and CD8_M respectively).”

As explained below, we believe that CD62L cannot be used to define naïve T cells in the SLOs of Foxo1^{TKO} mice: please see the CD62L fluorescence histograms shown in Figure S4 and the CD44/CD62L dot-plots below.

what LN were used as pLN?

Once again, we apologize for this omission. We have now indicated that cervical, axillary, brachial and inguinal peripheral lymph nodes (pLNs) were recovered and pooled in the “Methods” section of our revised manuscript (**lines 648-649**).

what do the colors of the correlogram signify in Fig. 5.

This point has now been clarified in the legend of the corresponding figure:

Lines 1169-1171: "Blue colors indicate positive correlations, while red colors indicate negative correlations. The darker the color, the more significant the correlation."

Of the specific major comments from the original critique, the points 2 (FMO, which should be shown in supplemental data), 4, 7, 8, 10, 11,12, 13, 15 and 16 are resolved. The other points (enumerated here per my original critique) still need to be answered:

FMO are now shown in supplemental Figures 1a (for Foxo1 and Foxo3) and Figures S2a (for pFoxo1 and pAKT).

1. I asked that the conventional Tn markers including CD62L and CCR7 be shown, appreciating that these may be reduced as a consequence of Foxo1 loss, but are not negative.

CD62L and CCR7 fluorescent histograms of naïve CD4 and CD8 T cells are now shown for representative young adult and old WT mice (Figure S2) as well as for young adult Foxo1^{TKO} and Foxo1^{Ctrl} mice (Figure S4). Representative dot-plots are also shown below. It is important to note that the extent of CD62L and CCR7 down-regulation is markedly more pronounced in naïve T cells from young adult Foxo1^{TKO} mice than in naïve T cells from old WT mice. More specifically, CD62L expression by naïve T cells from Foxo1^{TKO} mice includes not only low to intermediate levels of expression but also, for a significant proportion of cells, almost total absence of expression of this cell surface marker. Please note that our results are very similar to those published by Kerdiles et al. (Nat Immunol. 2009.10:176-84). As one of the objectives of this manuscript was to compare the T-cell compartment of old WT mice to that of young adult Foxo1^{TKO} mice, we have therefore chosen not to use CD62L to define naïve T cells.

I have also separately and elsewhere requested that CD49d (criticism #5) be included in the analysis to mark the Tvm cells (which are CD49d^{low}). The authors ignored the first request and stated that the second one could not be addressed because Foxo1 tightly correlated with CD49d expression – but my request was for the authors to show this analysis in wt old mice, and not Foxo1-TKO. Moreover, even for TKO cells, the figure provided in response to my question #5 shows expression (albeit lower) of CD49d in the spleen. This analysis is very important in understanding mechanisms of Foxo1-mediated changes in T cell migration with aging, and should be shown for all three molecules relative to Foxo1, and relative to one another and CD44, with CD49d differentiating between true and virtual memory;

Please, find below CD49d fluorescence histograms of CD4_M and CD8_M cells from the spleen of a representative old mouse and a representative young adult mouse (panel a). Percentages of CD49d⁻ cells among CD4_M, and CD8_M cells are also shown for the indicated SLOs of old/young adult mice (panel b). The proportion of CD49d⁻ cells is either decreased (in all SLOs for CD4_M cells and in the spleen for CD8_M cells) or unchanged (in pLNs and mLNs for CD8_M cells) with age. These results are in line with those previously published by Quinn et al. (Cell Reports. 2018. 23: 3512). Thus, the increase of memory T cell absolute numbers with age does not especially results from a preferential increase of virtual memory T cells (CD49d⁻ cells) at the expense of true memory T cells (CD49d⁺ cells).

Legends: Fluorescence histograms of CD4_M and CD8_M cells from the spleen of a representative old mouse and a representative young adult mouse (a). Percentages of CD49d⁻ cells among CD4_M and CD8_M cells are also shown for the indicated SLOs of old versus young adult mice (b).

As requested by the reviewer, we have also now quantified the percentages of CD49d⁻ cells among CD4_M and CD8_M cells from the SLOs of Foxo1^{Ctrl} and Foxo1^{TKO} mice (see the figure below). Surprisingly, by contrast to what we observed in old WT mice, we found that, with the exception of CD8_M cells from the spleen, the proportion of virtual memory (CD49d⁻) cells among CD4_M and CD8_M cells from the SLOs of Foxo1^{TKO} mice was significantly increased. These latter results suggest that full Foxo1 deficiency in T cells could promote the differentiation of naïve T cells into virtual memory T cells. So, as written by the reviewer, “while the TKO model is powerful to an extent, the parallels between a knockout and a physiological downregulation of a molecule are always limited.”

3. I accept the author’s response, and I caution them regarding the use of the term “inflammaging”, which has never been quantitatively or precisely defined by their original authors;

We thank the Reviewer for this helpful suggestion. The word “inflammaging” has now been removed from our revised manuscript.

6, and also 9. I have asked the authors to test whether the main effect of Foxo1 loss on Tn cells to downregulate CD62L and CCR7 and hamper their homing to LN. This question has not been addressed and is critical.

We have now performed the experiment suggested by the Reviewer. More precisely, we have compared the homing of purified Foxo1-deficient or Foxo1-sufficient CD4_N cells 18 hours after their transfer into recipient mice. In agreement with the results published by Kerdiles et al (Nat Immunol. 2009.10:176-84), we found that the ability of Foxo1-deficient CD4_N cells to home to lymph-nodes but not to the spleen was significantly impaired relative to that of Foxo1-sufficient CD4_N cells (Figure S4d,e).

Lines 201-210: "As previously described by Kerdiles et al. ²³, we observed that naïve T cells from Foxo1^{TKO} mice expressed low levels of CD62L and CCR7 compared to their T-cell counterparts from Foxo1^{ctrl} mice (Fig. S4b,c). As these 2 molecules are involved in the entry of T cells into lymph-nodes, their down-regulation may account for the decreased naïve T-cell cellularity observed in the lymph-nodes of Foxo1^{TKO} mice. To test this hypothesis, we next compared the homing of purified Foxo1-deficient or Foxo1-sufficient CD4_N cells 18 hours after their transfer into recipient mice (Fig. S4d). In agreement with Kerdiles et al. ²³, we found that the ability of Foxo1-deficient CD4_N cells to home to lymph-nodes but not to the spleen was significantly impaired compared with that of Foxo1-sufficient CD4_N cells (Fig. S4e)."

10. The enormous frequencies (50-100%) of T cells (particularly CD4_m and r) expressing PD-1 and also other exhaustion markers in old mice are reminiscent of true chronic infections and are not typical of either the literature (e.g. Decman, V. et al, J. Immunol., 2012; Renkema, K. et al., J. Immunol, 2014) or of our experience in SPF mice. The authors should provide the list of pathogens excluded from (and tested for) in their colony and also the levels of CXCL-10/IP-10 in their young and old mice, as well as in representative control mice purchased and analyzed immediately upon arrival. This is also important in interpreting the driving role of IFN- γ .

The latest health certificate for the mice of our animal house is attached at the end of this letter. Please note that all tests for rodent pathogens carried out during the five years we have been working on our aging colonies have been negative. It's worth noting that in the same room where we breed our colonies of old mice, we also have numerous strains of immunodeficient mice such as CD3 ϵ -KO mice and, to date, we haven't noticed any suspicious deaths in these colonies.

Concerning the 2 publications cited (Decman, V. et al, J. Immunol., 2012; Renkema, K. et al., J. Immunol, 2014) by the Reviewer:

- Decman et al.: They studied CD8 T-cell exhaustion in 18-month-old female C57BL/6 mice. They found 43% of PD1-positive cells among CD8_M cells in the spleen of their old mice (their Figure 1D). We found that 46.7% of CD8_M cells from the spleen of the 22-month-old female C57BL/6 mice of our colony expressed PD1 (our Figure 3c). Somehow, that is not so different, knowing that the mice we studied were 4 months older than the mice they studied.

- Benkema et al.: they found that around 20% of CD8 T cells from the spleen of 14-month-old OT-I TCR-transgenic mice expressed PD1 compared to less than 5% in 5-month-old mice (Their supplemental Figure 3). In our view, it is rather difficult to compare CD8_M cells from 22-month-old WT mice with the CD8 T cell pool (containing CD8_N cells) from 14-month-old OT-I TCR transgenic mice.

Finally, as suggested by the Reviewer, we measured the concentration of CXCL-10/IP-10 in the plasmas of young adult (3-month-old) and old (22-month-old) C57BL/6j mice from our animal facility as well as in the plasmas of 3-month-old C57BL/6j mice purchased from Janvier or Charles River and analyzed immediately upon arrival (see the figure below). On the one hand, IP-10 plasma levels do not differ between the 3 groups of young adult mice. On the other hand, IP-10 plasma levels are significantly higher in 22-month-old mice than in 3-month-old mice from the same colony bred at the Institut Cochin.

Legends: the concentration of CXCL-10/IP-10 in the plasma of young adult (3-month-old) and old (22-month-old) C57BL/6j mice from our animal facility as well as in the plasma of 3-month-old C57BL/6j mice purchased from Janvier or Charles River and

analyzed immediately upon arrival were measured using a kit from Meso Scale Discovery (MSD, V-PLEX Mouse IP-10 Kit - K152NVD-1).

Other points:

- Ln. 27, “are critical at the crossroad” should be replaced by “are critically involved at...”

Done

- Ln. 44, “the elderly” is considered a somewhat derogatory term by some individuals, and the recommended term is “older adults”

Done

- Ln. 53, only last names of authors should be in the citations (Kenyon, 2005)

Not all references were properly formatted: this has been corrected in the revised version of our manuscript.

- Ln. 276, ref. Crawford is missing.

Done

- IN light of inconsistent correlation between pAKT and pFoxo1 in Fig. 1, discussion on Ln. 461-67 needs to be softened and modified.

We agree with the reviewer. We have softened this part of the discussion in the revised version of our manuscript:

Lines 504-511: “Thus, taken together, our data suggest that the progressive decrease in Foxo1 expression with age may first result from chronic activation of the PI3K/AKT signaling pathway induced by inflammatory cytokines at least in CD4_N and CD8_N cells. This question remains open for the other T-cell subsets as we did not observe a clear correlation between AKT phosphorylation and Foxo1 levels in their cases. Secondly, as the transcription factor Foxo1 increases its own transcription^{60,61}, a decrease in Foxo1 expression at the protein level should ultimately lead to a decrease in Foxo1 transcription.”

Reviewer #2 (expert in inflamm-ageing and T cell function in old age):

The authors are applauded for their careful revisions. I do not have further comments.

We would like to thank the reviewer for his comment and we sincerely appreciate his enthusiasm.

Reviewer #3 (expert in Foxo transcription factors):

The authors partially addressed previously raised points. The study remains descriptive without solid mechanisms. For example, in Reviewer Points #6 and #10, - authors concluded that type 1 interferon-induced Foxo1 down-regulation involved the PI3K/Akt pathway. However, the authors' results show AKT inhibition further suppressed FOXO1 protein expression (Fig S9d). This contradicts conventional PI3K/AKT-mediated FOXO1 regulation. What is the molecular mechanism that regulates AKT-dependent FOXO1 expression in T cells?

First of all, we believe that reviewer 3 has misinterpreted our experiment, as we sincerely think that our results show that the PI3K/AKT signaling pathway is involved in the down-regulation of Foxo1 induced by type 1 interferons *in vitro*. Indeed, in the presence of inhibitors of PI3K and AKT, Foxo1 down-regulation in response to IFN α 4 was either completely abolished (in all T-cell subsets for the inhibitor of PI3K and in CD4 T cells for the inhibitor of AKT) or at least significantly reduced (in CD8 T cells for the inhibitor of AKT).

However, we realized that our data representation was misleading. In the previous version of our manuscript, for each T-cell subset, each individual graph was related to a given inhibitor. Therefore, we were comparing Foxo1 relative MFI after 4 days of culture in the presence of IFN α 4 and a given inhibitor to the Foxo1 relative MFI when cultured with the same inhibitor alone. So, the statistics presented at that time only indicated whether or not type 1 interferons still induce a significant down-regulation of Foxo1 in the presence of the indicated inhibitor. These statistics did not allow the reader to estimate whether the extent of Foxo1 down-regulation induced by IFN α 4 was reduced in the presence of the inhibitors tested. We sincerely apologize, as we have now realized that our graphs were very confusing.

In the revised version of our manuscript, we have chosen a different way of representing our data (a way very similar to the one used for Fig. 6d) in which the statistics allow to directly compare IFN α 4-driven changes in the expression of Foxo1 when added in the culture medium alone or together with the inhibitors tested (see the Figures 7b and S10d of our revised manuscript). More precisely, the data have been normalized exactly the same way than in our previous version, i.e. Foxo1 MFI in the presence of IFN α 4 + a given inhibitor was divided by the Foxo1 MFI in the presence of the same inhibitor alone (for these 2 conditions, Foxo1 staining was performed in a single well after labelling the 2 cell suspensions with anti-CD45 antibodies conjugated to different fluorochromes). We have only chosen, for each T-cell subset, to plot the condition "IFN α 4 alone" together with the conditions "IFN α 4 + the various tested inhibitors". As a result, the new figures (Figures 7b and S10d) are in fact very similar to the previous ones, except that they present the different groups directly side by side. This representation allows to easily compare whether the tested inhibitors were decreasing the extent of IFN α 4-induced Foxo1 down-regulation. Thus, we hope that our revised figures now clearly show that PI3Ki and AKTi significantly reduced the extent of Foxo1 down-regulation induced by IFN α 4 in all T-cell subsets and again, we apologize for the confusion.

The text of our manuscript was modified accordingly:

Lines 436-440: *"To do so, we cultured purified T cells in the presence of IFN- α alone or with several inhibitors of key kinases for 4 days. (Fig. 7a). As expected, inhibitors of TYK2 and JAK1, 2 kinases directly associated with the 2 chains of the type 1 interferon receptor completely abolished (CD4 T-cell subsets) or reduced (CD8 T-cell subsets) type 1 interferon-induced down-regulation of Foxo1 (Fig. 7b and Fig. S10d)."*

Lines 445-448: *"Indeed, in the presence of inhibitors of PI3K and AKT, Foxo1 down-regulation in response to IFN- α was either completely abolished (in all T-cell subsets for the inhibitor of PI3K and in CD4 T cells for the inhibitor of AKT) or at least significantly reduced (in CD8 T cells for the inhibitor of AKT)."*

In addition, how viable are these CD8 T cells following treatment with inhibitors? Downregulation of FOXO1 following AKTi indicates cells have lost viability. Measuring FOXO1 expression in sick cells will surely show downregulation, regardless of actual regulation.

All the inhibitors we used were first titrated in order to find the highest concentration we could use without killing T cells. To do so, purified T cells were cultured for 4 days with IL7 alone or IL-7 + various concentrations of the inhibitor tested. More precisely, to properly determine and compare the mortality induced by the inhibitors tested after culture, a similar number of wells after a 4-day culture in the presence or absence of a given inhibitor were mixed after labeling the cells with anti-CD45 antibodies conjugated to different fluorochromes (PE-Cy7 for the tested inhibitor + IL7 versus BUV395 for IL-7 alone), so that staining was performed in a single well. Then, we determined the proportion of CD45- PE-Cy7-labelled cells after staining (see the examples below shown for the inhibitors of AKT and PI3K). When the proportion of cells that were cultured with an inhibitor was 50%, this indicates that at this concentration, the added inhibitor did not induce more cell death of cultured cells than when these cells were cultured with IL-7 alone. On these 2 graphs, the concentration finally chosen for the experiments described in Figures 7b and S10d is indicated.

Legends: Purified T cells were cultured for 4 days with IL7 alone or IL-7 + AKTi (left panel) or PI3Ki (right panel). After 4 days, a similar number of wells in the presence or absence of the indicated inhibitor were mixed after labeling the cells with anti-CD45 antibodies conjugated to different fluorochromes (PE-Cy7 for the tested inhibitor + IL7 versus BUV395 for IL-7 alone), so that staining was performed in a single well. Then, the proportion of CD45- PE-Cy7-labelled cells was then determined. The dashed box indicates the inhibitor concentration chosen for subsequent experiments.

Aside from this confusing result, the authors do not delineate the molecular mechanism or signaling pathway that results in FOXO1 expression in aging T cells.

Once again, we believe that our way of representing our results was misleading, leading to some confusion. We would like to sincerely apologize for this and hope that our new Figures 7b and S10d are now more convincing. We do believe that our data strongly suggest that the PI3K/AKT signaling pathway is involved in the down-regulation of Foxo1 induced by type 1 interferons *in vitro*.

This manuscript requires a careful editing. For example, in lines 466-467 the sentence does not make good sense.

This sentence has been rewritten and the manuscript edited.

REVIEWERS' COMMENTS

Reviewer #1 (Remarks to the Author):

The manuscript by Durand et al. has again been improved via revision. The manuscript outlines links between FOXO1, pAKT, IFN-I and age-related changes in some T cell subsets. The authors have done a better job to carefully place limits on such links and outline where they do not appear to meet biological and/or statistical significance. As they have done so, I have found it difficult to really get excited about the priority of discoveries made here. Overall, the authors have shown that IFN-I contributes to the above changes to some extent, and that pAKT and FOXO1 are somewhat involved in the same processes. They may be sometimes on the same axis, sometimes they are not, and each, and particularly IFN-I, may have independent effects. These changes apply to some, but not other, memory T cell subsets. For sure, biology is often, if not always, complex. However, this story remains decidedly muddled. It is ultimately the matter of editorial decision on whether this manuscript represents a coherent significant contribution whose priority meets publication in Nature Communications.

Reviewer #3 (Remarks to the Author):

The authors have addressed my comments with thoughtful consideration of the feedback provided.